# ZBTB18 inhibits SREBP-dependent lipid synthesis by halting CTBPs and LSD1 activity in glioblastoma

Roberto Ferrarese[1],*, Annalisa Izzo[1],*, Geoffroy Andrieux[2,3],*, Simon Lagies[4,5,6], Johanna Paulina Bartmuss[1], Anie Priscilla Masilamani[1], Alix Wasilenko[1], Daniela Osti[7], Stefania Faletti[7], Rana Schulzki[1], Shuai Yuan[1], Eva Kling[1], Valentino Ribecco[1], Dieter Henrik Heiland[1,3], Stefan Tholen[8], Marco Prinz[9,10,11], Giuliana Pelicci[7,12], Bernd Kammerer[4,5,13], Melanie Boerries[2,3], Maria Stella Carro[1]

**Enhanced fatty acid synthesis is a hallmark of tumors, including glioblastoma. SREBF1/2 regulate the expression of enzymes involved in fatty acid and cholesterol synthesis. Yet, little is known about the precise mechanism regulating SREBP gene expression in glioblastoma. Here, we show that a novel interaction between the co-activator/co-repressor CTBP and the tumor suppressor ZBTB18 regulates the expression of SREBP genes. In line with our findings, metabolic assays and glucose tracing analysis confirm the reduction in several phospholipid species upon ZBTB18 expression. Our study identifies CTBP1/2 and LSD1 as co-activators of SREBP genes and indicates that the functional activity of the CTBP-LSD1 complex is altered by ZBTB18. ZBTB18 binding to the SREBP gene promoters is associated with reduced LSD1 demethylase activity of H3K4me2 and H3K9me2 marks. Concomitantly, the interaction between LSD1, CTBP, and ZNF217 is increased, suggesting that ZBTB18 promotes LSD1 scaffolding function. Our results outline a new epigenetic mechanism enrolled by ZBTB18 and its co-factors to regulate fatty acid synthesis that could be targeted to treat glioblastoma patients.**

## Introduction

With a median survival rate of 15 mo and a 5-yr overall survival rate of 5.5%, glioblastoma (GBM) is the most aggressive and deadly type of brain tumor (Ostrom et al, 2015). Over the years, GBM eluded even the most aggressive treatments (surgery followed by chemo- and radiotherapy), largely because of the invasiveness and chemo/radio-resistance of the residual tumor cells that escape resection and the subsequent treatment (Wang et al, 2016a).

GBM tumors with a mesenchymal phenotype have been described as the most aggressive, possessing features such as invasion and therapy resistance (Phillips et al, 2006; Verhaak et al, 2010; Bhat et al, 2013; Wang et al, 2017). The acquisition of mesenchymal traits in GBM is reminiscent of the mesenchymal transition in epithelial tumors, and it is now considered a hallmark of aggressive tumors (Fedele et al, 2019). Epigenetic changes in the cells are inheritable, reversible covalent modifications altering gene expression without changing the DNA sequence. Histone-modifying enzymes control this process by adding or removing acetyl or methyl groups to or from specific positions of these proteins and thus regulating chromatin accessibility to the transcriptional machinery. These enzymes usually interact with transcription factors, co-repressors, or co-activators and represent an important therapeutic opportunity in cancer (Romani et al, 2018).

The C-terminal binding proteins (CTBP1/2) can function as co-repressors through association with DNA-binding transcription factors and recruitment of chromatin regulators such as histone deacetylases 1 and 2 (HDAC1/2), lysine-specific demethylase 1 (KDM1A/LSD1), the histone methyltransferase PRC2, and the chromatin remodeling complex NURD (Shi et al, 2003; Boxer et al, 2014; Kim et al, 2015). Although usually described as repressors, CTBP1/2 have been also reported to act as co-activators through the interaction with retinoic acid receptors (Bajpe et al, 2013) or through binding to the zinc finger protein RREB1 (Ray et al, 2014). CTBP1/2 have been also implicated in tumorigenesis and epithelial-to-mesenchymal transition (EMT) (Di et al, 2013). In GBM, CTBP1/2

[1]Department of Neurosurgery, Medical Center–University of Freiburg, Faculty of Medicine, University of Freiburg, Breisgau, Germany   [2]Institute of Medical Bioinformatics and Systems Medicine, Medical Center–University of Freiburg, Faculty of Medicine, University of Freiburg, Freiburg, Germany   [3]German Cancer Consortium (DKTK), Partner Site Freiburg and German Cancer Research Center (DKFZ), Heidelberg, Germany   [4]Center for Biological Systems Analysis, University of Freiburg, Breisgau, Germany   [5]Spemann Graduate School of Biology and Medicine (SGBM), University of Freiburg, Freiburg, Germany   [6]Faculty of Biology, University of Freiburg, Freiburg, Germany   [7]Department of Experimental Oncology, IEO, European Institute of Oncology, IRCCS, Milan, Italy   [8]Institute of Clinical Pathology, Medical Center-University of Freiburg, Faculty of Medicine, University of Freiburg, Freiburg, Germany   [9]Institute of Neuropathology, Medical Center-University of Freiburg, Faculty of Medicine, University of Freiburg, Freiburg, Germany   [10]Signaling Research Centres BIOSS and CIBSS, University of Freiburg, Freiburg, Germany   [11]Center for NeuroModulation (NeuroModul), University of Freiburg, Freiburg, Germany   [12]Department of Translational Medicine, Piemonte Orientale University "Amedeo Avo-Gadro," Novara, Italy   [13]BIOSS Centre of Biological Signaling Studies, University of Freiburg, Freiburg Germany

Correspondence: maria.carro@uniklinik-freiburg.de
*Roberto Ferrarese, Annalisa Izzo, and Geoffroy Andrieux contributed equally to this work.

have been shown to be highly expressed compared with lower grade tumors and to be associated with poorer prognosis (Wang et al, 2016b).

KDM1A/LSD1 is a histone demethylase, which acts as a co-repressor or co-activator depending on the function of its protein complex. LSD1 mostly removes mono- and di-methylation of histone H3 at lysine K4 (H3H4me1/2) (Shi et al, 2004). Demethylation of H3K9me2, a transcriptional repression marker of heterochromatin, has also been reported to be an LSD1 target, especially upon interaction with nuclear receptors (Metzger et al, 2005; Garcia-Bassets et al, 2007).

Sterol regulatory element–binding proteins (SREBPs) are transcription factors that control the expression of enzymes involved in fatty acid and cholesterol biosynthesis (Horton et al, 2002). SREBF1 regulates fatty acid synthesis, whereas SREBF2 is implicated in cholesterol production (Horton et al, 2002). Fatty acid synthesis plays an important role in cancer, including GBM (Bensaad et al, 2014; Lewis et al, 2015); an excess of lipids and cholesterol in tumor cells are stored in lipid droplets (LD), a hallmark of cancer aggressiveness (Menendez & Lupu, 2007). In GBM, constitutive activation of EGFR (EGFRvIII mutant) leads to PI3K/AKT-dependent SREBF1 regulation with a consequent increase in lipogenesis and cholesterol uptake, which can be pharmacologically blocked by inhibiting the low-density lipoprotein receptor (LDLR) (Guo et al, 2011). More works have further established a role of SREBF1 as a promoter of GBM growth (Cheng et al, 2015; Geng et al, 2016; Ru et al, 2016). Blocking LD formation suppresses GBM lipogenesis and growth (Geng et al, 2016); furthermore, the activation of the SREBP pathway has been recently connected to the mesenchymal shift in GBM (Schmitt et al, 2021).

Previously, we have identified ZBTB18 as tumor suppressor, which is low expressed in GBM and GBM cell lines. Others and we have shown that ZBTB18 functions as a transcriptional repressor of mesenchymal genes and impairs tumor formation (Carro et al, 2010; Tatard et al, 2010; Fedele et al, 2017; Xiang et al, 2021). Here, we identify CTBP1 and CTBP2 as new ZBTB18-binding proteins. We report that CTBP and LSD1 transcriptionally activate the expression of fatty acid synthesis genes and that such activation is opposed by ZBTB18 through the inhibition of LSD1-dependent demethylase activity.

# Results

### ZBTB18 interacts with CTBP through the VLDLS motif

With the goal to get a better insight into ZBTB18 transcriptional repressive mechanisms, we used mass spectrometry (MS) to identify ZBTB18 co-precipitated proteins in glioblastoma cells (SNB19), upon FLAG-ZBTB18 overexpression and subsequent anti-FLAG co-immunoprecipitation (Fig S1A). Among other proteins, CTBP1 and CTBP2 appeared to be potential ZBTB18 interactors (Fig S1B). We decided to focus on these proteins given their role as cofactors in gene expression regulation and connection to EMT and cancer (Di et al, 2013; Wang et al, 2016b). We validated the interactions by co-immunoprecipitation in SNB19 cells expressing ectopic ZBTB18

using different antibodies directed to ZBTB18 or CTBP2 (Fig 1A and B). CTBP1, on the contrary, appeared to bind to CTBP2 but not to ZBTB18 directly (data not shown). We also confirmed ZBTB18 interaction with CTBP2 in LN229 cells transduced with ZBTB18 (Fig S1C and D). Here, the amount of endogenous CTBP2 was not enough to be detected in the 2% of the input but appeared in the precipitated fraction, and its identity was confirmed by comparison with the band detected in the total protein lysate (Fig S1C and D). To prove that the endogenous ZBTB18 is also able to interact with CTBPs, co-IP followed by MS was performed in the GBM-derived brain tumor stem cell (BTSC) line, which naturally expresses ZBTB18 (BTSC268) (Figs 1C and S1E) (Masilamani et al, 2022). The interaction was further corroborated in a second BTSC line (BTSC475) with a basal level of ZBTB18 (Fig S1F) (Masilamani et al, 2022). Although not sufficient to be detected in the 2% of the input, in both BTSCs the endogenous ZBTB18 appeared in the precipitated fraction with CTBP (Figs 1C and S1F). Furthermore, co-IP in U3082 described in our recent study also highlighted ZBTB18 binding to CTBP (Masilamani et al, 2022); here, CTBP1, but not CTBP2, was detected in the co-precipitated fraction (Fig S1G), suggesting that ZBTB18 preferential interaction with CTBP2 or CTBP1 could be context-dependent. ZBTB18 protein sequence analysis revealed the presence of a putative CTBP interaction motif (VLDLS) (Nibu & Levine, 2001). Substitution of the Leu/L240, Asp/D241, and Leu/L242 amino acids into Ser/S, Ala/A, and Ser/S by site-directed mutagenesis in SNB19 cells completely abolished CTBP2 interaction with ZBTB18 (Fig 1D and E), further validating ZBTB18-CTBP interaction. Consistent with our previous study (Fedele et al, 2017), ZBTB18 affected cell proliferation, apoptosis, and migration (Fig S2A–F). ZBTB18 LDL mutant (ZBTB18-mut) also appeared to have a mild effect on apoptosis and proliferation, whereas migration was not impaired (Fig S2A–F). Expression analysis of previously validated ZBTB18 mesenchymal targets, upon overexpression of the ZBTB18-mut, showed that ZBTB18 interaction with CTBP is required for ZBTB18-mediated repression of a subset of targets (CD97, LGALS1, and S100A6; Fig 1F), whereas repression of ID1, SERPINE1, and TNFAIP6 type of genes appears to be independent from CTBP2 binding. Overall, these data identify CTBP1/2 as new ZBTB18 interactors in GBM cells and suggest that ZBTB18 might employ both a CTBP-dependent and a CTBP-independent mechanism to repress target genes and mediate its tumor suppressor functions.

### ZBTB18 and CTBP2 play an opposite role in gene expression regulation

Because CTBP1/2 are mostly known as co-repressors, we hypothesized that CTBP1/2 could be required for the ZBTB18 function. To verify this possibility, we overexpressed FLAG-ZBTB18 and knocked down CTBP2, which interacts with ZBTB18, in SNB19 cells (Fig 2A). Gene expression profiling followed by gene set enrichment analysis (GSEA) showed that both ZBTB18 overexpression and CTBP2 silencing similarly affected the expression of EMT gene signatures, suggesting an opposite role of CTBP2 and ZBTB18 in transcriptional regulation (Fig S3A). This is consistent with our reported role of ZBTB18 as an inhibitor of mesenchymal signatures in GBM and with previous studies indicating that CTBP2 promotes tumorigenesis and EMT.

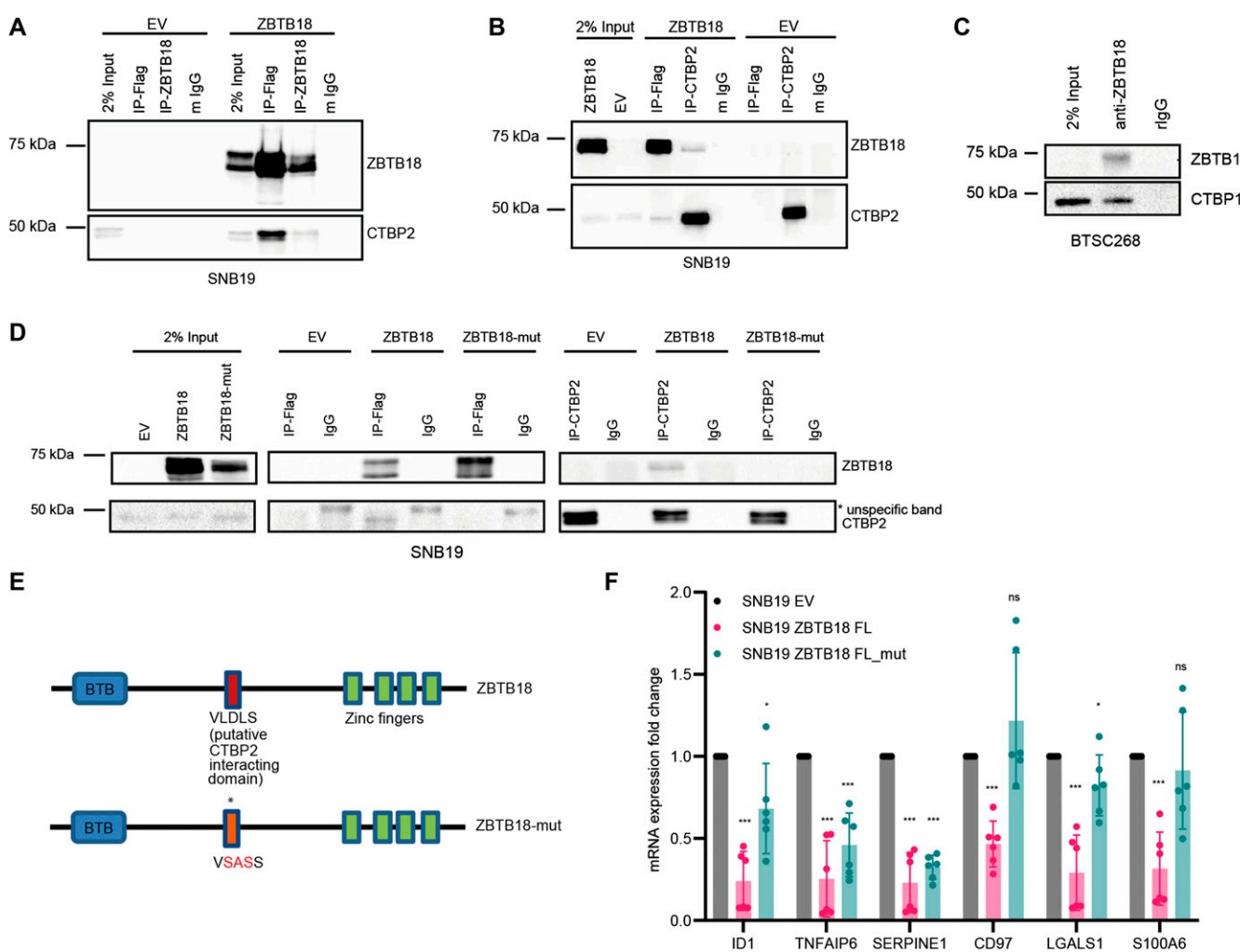

**Figure 1. ZBTB18 interacts with CTBP through the VLDLS domain in glioblastoma cells.**
**(A, B)** Western blot (WB) analysis of FLAG and ZBTB18 co-IP (A) or FLAG and CTBP2 co-IP (B) in SNB19 cells transduced with EV or FLAG-ZBTB18. **(C)** WB analysis of endogenous ZBTB18 co-IP in BTSC268 primary glioblastoma cells. **(D)** WB analysis of FLAG co-IP (middle panel) or CTBP2 co-IP (right panel) in SNB19 cells expressing either FLAG-ZBTB18 or FLAG-ZBTB18-mut. **(E)** Schematic representation of the ZBTB18 protein with BTB and zinc finger domains. The putative CTBP2-interacting motif (VLDLS) and the corresponding mutation are marked. **(F)** qRT–PCR showing the expression of ZBTB18 targets upon ZBTB18 and ZBTB18-mut overexpression in SNB19 cells. n = 3 biological replicates; error bars ± s.d. *P < 0.05, **P < 0.01, and ***P < 0.001 by a t test. Gene expression was normalized to 18S RNA.

Venn analysis of shCTBP2- and ZBTB18-regulated genes showed that about half of the genes regulated by shCTBP2 are also repressed upon ZBTB18 overexpression (Fig 2B, top panel). CTBP2 knockdown partially affected the portion of ZBTB18-regulated genes supporting the observation that ZBTB18 may regulate gene expression both in a CTBP2-dependent and in a CTBP2-independent manner (Fig 2B, bottom panel). GSEA revealed a strong loss of SREBP signaling gene expression, upon both ZBTB18 overexpression and CTBP2 silencing (Figs 2C and D and S3B), according to a possible role of CTBP2 as a co-activator and of ZBTB18 as a repressor. SREBP genes are involved in fatty acid synthesis; we focused on this pathway given its importance in glioblastoma tumorigenesis and previous connection to EMT (Guo et al, 2011; Cheng et al, 2015; Wang et al, 2015; Yang et al, 2015; Geng et al, 2016; Ru et al, 2016; Zhang et al, 2019). Interestingly, SREBF1 activation has been recently shown to promote a mesenchymal

shift in GBM (Schmitt et al, 2021). Consistent with the literature reports, GlioVis analysis (Bowman et al, 2017) showed that SREBF1 is more expressed in the most aggressive GBM subtypes (mesenchymal and classical) and correlates with patient survival (Fig S3C and D). Furthermore, we observed a strong positive correlation between CTBP2 expression in gliomas and the levels of all the SREBP genes tested (GlioVis platform, dataset CGGA [Zhao et al, 2017 and Fig S4]), which is in line with a possible role of CTBP2 as a co-activator of SREBP genes. Validation by qRT-PCR in SNB19 cells confirmed the reduction in the expression of most of the genes tested, upon ZBTB18 expression or CTBP2 silencing (Fig 2E). In agreement with our microarray results, CTBP2 knockdown did not inhibit ZBTB18-mediated repression (Fig 2D and E), but rather it further enhanced the down-regulation of ZBTB18 target genes. Remarkably, treatment with the CTBP inhibitor MTOB (Achouri et al, 2007) also resulted in a strong down-regulation of many SREBP

**Figure 2. ZBTB18 and CTBP regulate the expression of sterol regulatory element–binding protein (SREBP) genes involved in fatty acid synthesis.**
**(A)** WB analysis of FLAG-ZBTB18 and CTBP2 expression in SNB19 cells transduced with either FLAG-ZBTB18 or shCTBP2. **(B)** Venn analysis showing the overlap between regulated genes in ZBTB18 versus empty vector (EV) and genes regulated in shCTBP2 versus shCtr (upper part) or in ZBTB18 shCTBP2 versus EV (lower part). Genes were selected based on adjusted *P* < 0.05 and absolute fold change > 0.5. **(C)** Top 10 down-regulated consensus pathways in the overlap between shCTBP2- versus shCtr- and between ZBTB18- versus EV-regulated genes (ZBTB18 versus shCTBP2) from Fisher's exact test comparing the DEGs (adj. *P* < 0.05, FC < −0.5) with the whole set of quantified genes. Processes related to SREBP signaling are highlighted. **(D)** Row-wise z-score heatmap showing the expression of SREBP signaling genes in each triplicate across the four conditions. Both row and column hierarchical clustering are based on the Euclidean distance, and the complete clustering method was used. **(E)** qRT-PCR

genes in SNB19 cells (Fig 2F). We then sought to confirm our findings upon ectopic expression of ZBTB18 in BTSC233 and BTSC168 lines, which do not express ZBTB18, as previously shown (Figs 2G and H and S5A and B) (Masilamani et al, 2022). Treatment with MTOB in BTSC233 led to the down-regulation of many SREBP genes, especially SREBF1 (Fig 2H). Of note, when MTOB treatment was combined with ZBTB18 or ZBTB18-mut expression, no rescue of ZBTB18-mediated repression was observed (Fig 2H). In fact, in some cases (SREBF1, GPAM, SQLE, and SCD) the combined effect of MTOB and ZBTB18 appeared to be even stronger than the single treatments, similar to what we had observed when ZBTB18 was expressed in concomitance with CTBP2 knockdown (Fig 2E). In the presence of ZBTB18-mut, which has no significant effect on SREBF1, MTOB nonetheless elicited repression of the gene expression (Fig 2H); this is consistent with the idea that ZBTB18 and MTOB independently impair CTBP-mediated activation. However, although less efficient than ZBTB18 in repressing some of the tested SREBP genes, ZBTB18-mut still showed some degree of gene down-regulation. This suggests that ZBTB18 may also be able to repress gene expression independent from CTBP binding, as previously observed (Fig 1H). Knockdown of CTBP1 and CTBP2, either alone or in combination, in BTSCs further proved CTBPs' activating role (Figs 2I and J and S5C–E). These results corroborate the idea that ZBTB18 represses CTBP2, which in turn functions as an activator of SREBP genes. We then attempted to knock down ZBTB18 in BTSC475 cells, which express ZBTB18 (Masilamani et al, 2022), by CRISPR/Cas9 (Fig S6A and B). However, only a modest re-expression of LDLR and SCD was detected, probably as a consequence of the low expression level of ZBTB18 (Fig S6C).

## ZBTB18 affects lipid synthesis and reduces lipid storage

Then, we investigated whether ZBTB18 overexpression caused phenotypic changes associated with the reduction in fatty acid synthesis, because of the deregulated expression of SREBP genes. SNB19 cells were transduced with ZBTB18 or ZBTB18-mut, and profiling of lipid species expression was performed by a targeted LC/MS method (Fig 3A). Hierarchical clustering based on lipid species relative abundance highlighted that ZBTB18-expressing cells segregated together with each other and separately from the other samples (empty vector [EV] and ZBTB18-mut) regardless of the growing conditions (normal medium or lipid-depleted medium) (Fig 3B and C). Within this cluster, significantly regulated lipids were largely underrepresented in the ZBTB18-expressing cells compared with the EV control cells; notably, the lipid starvation exacerbated this trend.

To further confirm a role of ZBTB18 in lipid turnover in GBM cells, we measured lipid droplet abundance upon ZBTB18 overexpression. In both the tested primary GBM stem cell lines (BTSC168 and BTSC233), the presence of ZBTB18 led to a significant reduction in the amount of lipid droplets within the cells (Fig 3D–F). A similar effect was observed in SNB19 cells, in which the reduced number of lipid droplets upon ZBTB18 overexpression became more evident after 48-h lipid starvation (Fig S7A and B). Moreover, when lipid-starved, ZBTB18-expressing cells were incubated with a lipid-containing medium again, they showed a significant increment in lipid droplets albeit not fully recovering to the level of the EV controls (Fig S7A and B). This observation suggests that the loss of lipid droplets in cells expressing ZBTB18 is mostly due to the blockade of the lipid biosynthesis on which tumor cells especially rely when there are no lipid sources available in the environment. In agreement with this hypothesis, no significant difference in the uptake of fluorescently labeled palmitic acid (Bodipy-C16) was observed between ZBTB18-expressing cells and the respective controls (Fig S7C). Then, we measured lipid droplet abundance upon ZBTB18 knockdown in BTSC475; here, despite the modest knockdown, the number of lipid droplets significantly increased upon ZBTB18 deletion with two sgRNAs (Fig 3G and H), further supporting our data. To further estimate the impact of ZBTB18 on de novo lipogenesis, we set up a $^{13}$C incorporation assay using labeled glucose in BTSC168 to quantitatively measure fatty acid synthesis. The results showed that, upon expression of ZBTB18, the detected fatty acids contained comparatively less glucose-derived $^{13}$C isotopes, suggesting a diminished request of carbon atoms for de novo lipogenesis (Fig 3I). Taken together, these results suggest that ZBTB18 plays an important role in controlling lipid metabolism of GBM cells.

## ZBTB18 and CTBP2 map to SREBP promoter regions

To characterize the dynamics between CTBP2 and ZBTB18 on gene target regulation, we mapped the genome-wide distribution of CTBP2 and ZBTB18 by chromatin immunoprecipitation coupled to deep sequencing (ChIP-seq) in SNB19 cells. CTBP2 ChIP was performed with a CTBP2 antibody to precipitate the endogenous CTBP2 in the absence or in the presence of ZBTB18 (EV_ChIP-CTBP2 and ZBTB18_ChIP-CTBP2), whereas ectopic ZBTB18 was immunoprecipitated with FLAG antibody (EV_ ChIP-FLAG and ZBTB18_ ChIP-FLAG) (Fig 4A).

After subtracting FLAG-unspecific peaks detected in SNB19-EV_ChIP-FLAG, we performed a Venn analysis of all the annotated peaks. We found 5,361 peaks shared by all conditions, suggesting that CTBP2 and ZBTB18 bind to the same genomic regions (Fig 4B). Remarkably, peaks that are in common with all the conditions (All)

validation of selected SREBP gene expression upon ZBTB18 expression and/or CTBP2 silencing in SNB19 cells. n = 3 biological replicates; error bars ± s.d. *P < 0.05, **P < 0.01, and ***P < 0.001 by a t test. Gene expression was normalized to 18S RNA. **(F)** qRT–PCR analysis of selected SREBP genes in SNB19 cells treated with the CTBP2 inhibitor MTOB. n = 3 biological replicates; error bars ± s.d. *P < 0.05, **P < 0.01, and ***P < 0.001 by a t test. Gene expression was normalized to 18S RNA. **(G)** WB analysis of ZBTB18 expression in BTSC233 cells transduced with empty vector (EV), ZBTB18, or ZBTB18-mut, using a ZBTB18 antibody. **(H)** qRT–PCR analysis of selected SREBP genes upon ZBTB18 overexpression and MTOB treatment. Results are presented as the mean of n = 3 biological replicates; error bars ± s.d. *P < 0.05, **P < 0.01, and ***P < 0.001 by a t test. Gene expression was normalized to 18S RNA. **(I)** Western blot analysis of CTBP1 (top panel) and CTBP2 (bottom panel) expression upon silencing in BTSC233 cells. **(J)** qRT–PCR analysis of SREBP targets in BTSC233 transduced with shCTBP1- or shCTBP2-expressing lentivirus. Two independent shRNAs to knock down CTBP1 or CTBP2 were used. Results are presented as the mean of n = 3 biological replicates; error bars ± s.d. *P < 0.05, **P < 0.01, and ***P < 0.001 by a t test. Gene expression was normalized to 18S RNA.

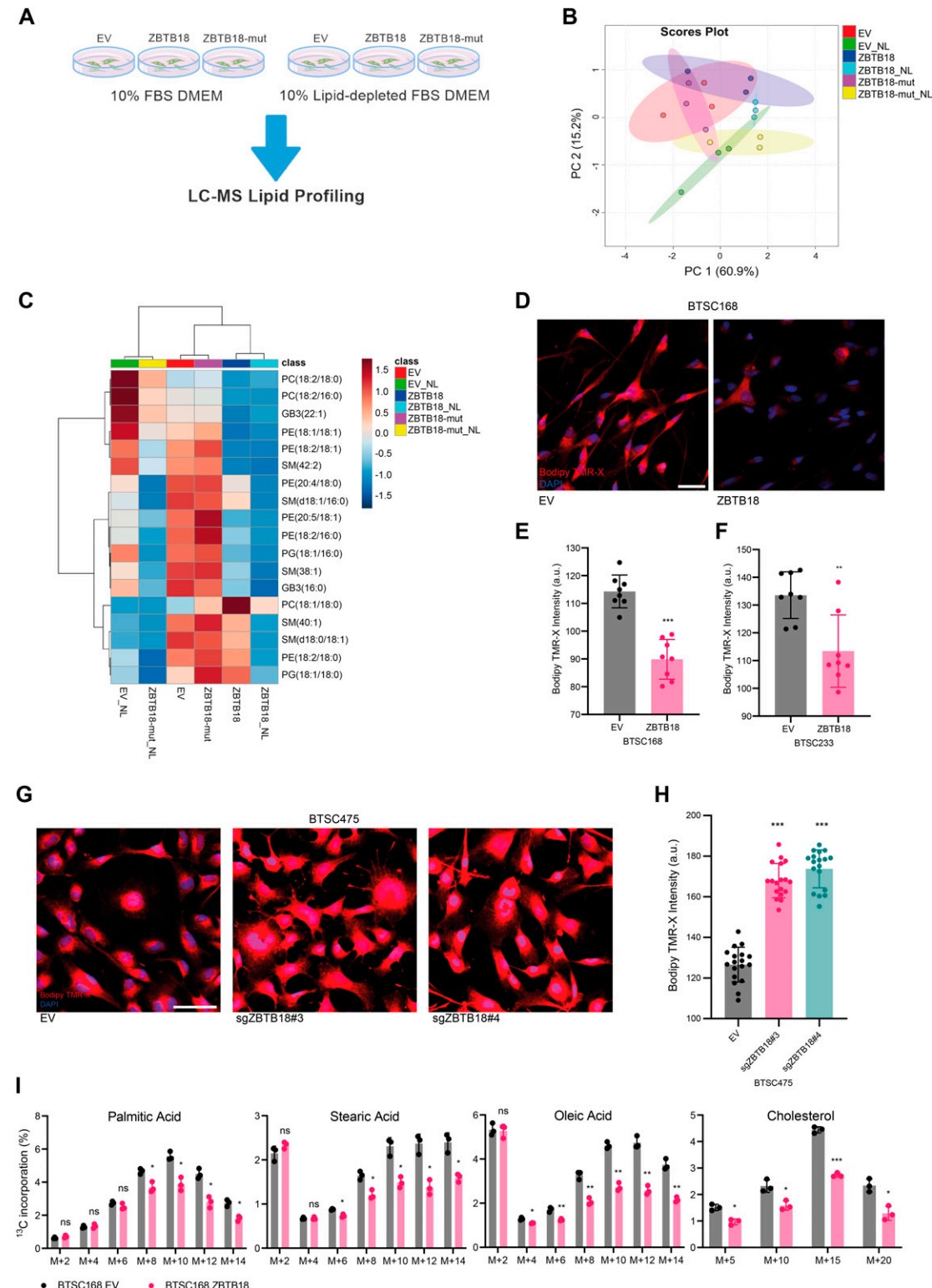

**Figure 3. ZBTB18 overexpression alters lipid content depending on its CTBP binding capability.**
**(A)** Experimental flowchart of lipidomics analysis. **(B)** Principal component analysis of identified lipids in SNB19 cells transduced with empty vector (EV), ZBTB18-overexpressing vector (ZBTB18), or ZBTB18 mutant (ZBTB18-mut) and grown in the presence (no suffix) or absence of lipids (NL). n = 3 biological replicates. **(C)** Heatmap of significantly altered (q < 0.05) lipids in SNB19 cells expressing EV, ZBTB18, or ZBTB18-mut, grown in the presence (no suffix) or absence of lipids (NL). Range-scaled z-scores are displayed. **(D)** Bodipy TMR-X lipid staining of BTSC233 cells expressing EV or ZBTB18. Nuclei were counterstained with DAPI. Scale bar: 100 μm. **(D, E, F)** Quantification of Bodipy TMR-X lipid staining in BTSC168 (D, E) and BTSC233 (F). n = 4 biological replicates. Five images for each of the four biological replicates were taken with a confocal

and those that are shared between ZBTB18 ChIP and CTBP2 ChIP when ZBTB18 is expressed (Not EV_CTBP2) were strongly enriched at promoter regions in close proximity to the transcription start (Fig 4C and D). This suggests that although CTBP2 and ZBTB18 share common sites (All), a fraction of CTBP2 can be recruited to new promoter regions when ZBTB18 is expressed (Not EV_CTBP2). In silico analysis of consensus binding motifs by HOMER showed enrichment for ZBTB18 in both groups (Not EV CTBP2 and All), in addition to the "only ZBTB18_FLAG" group (Fig 4E). Promoter peaks belonging to the group "Not ZBTB18_CTBP2," which are not shared by ZBTB18 and CTBP2 upon ZBTB18 expression, do not contain the ZBTB18 motif, suggesting that these peaks could be unspecific (Fig 4E). Analysis of published ChIP-seq datasets using ReMap (Cheneby et al, 2020) showed a good overlap between shared CTBP2 and ZBTB18 promoter peaks, with peaks for known CTBP2 interactors such as NCOR1, ZNF217, and LSD1/KDM1A ($P$ = 0; Fig S8A and C). When focusing on SREBP gene promoters, we observed a strong consensus between peaks identified in ZBTB18 (FLAG) and CTBP2 ChIP (Fig 4F), again suggesting that CTBP2 and ZBTB18 play a direct role in SREBP gene transcription. Furthermore, SREBP gene peaks common to ZBTB18 and CTBP2 ChIP matched with genes regulated by CTBP2 silencing and ZBTB18 overexpression in our gene expression analysis (*INSIG1*, *SREBF1*, *FASN*, *ACACA*, *LDLR*, *DBI*, *SCAP*, *SQLE*, and *SCD*; Figs 2D and E and 4G). Furthermore, ReMap analysis revealed a good consensus between regions shared by ZBTB18 and CTBP2 and SREBP peaks ($P$ = 0; Fig S8B and C). Together, these results further reinforce the notion that CTBP2 and ZBTB18 bind to common gene promoters, which include SREBP genes. In addition, this constitutes the first genome-wide mapping of CTBP2 and ZBTB18 in GBM cells.

We further confirmed the binding of ZBTB18 to the promoters of a set of SREBP genes by qChIP analysis in glioblastoma cells. Upon overexpression in SNB19 cells, ZBTB18 is recruited to the promoter region of all analyzed genes but not of *CYP51A1*, which is a preferential target of CTBP2 alone (Fig 4H). CTBP2 is present at the promoter of *SREBP* genes in the absence of ZBTB18 (Fig 4I) where according to our microarray analysis and qRT-PCR results it is responsible for their transcriptional activation. In line with the proposed role of ZBTB18 as a repressor of SREBP gene expression, we observed that the levels of H3K4me3, a well-known marker of transcriptional activation, decreased at the promoters of all the analyzed genes (Fig S9A). Conversely, the level of the repressive marker H3K9me2 increased (Fig S9B).

### ZBTB18 facilitates CTBP2 binding to ZNF217 and LSD1

To better understand whether the presence of ZBTB18 affects CTBP2 interaction with other proteins, we performed SILAC-based MS in SNB19 cells transduced with either control vector (low-weight medium, L) or ZBTB18 (high-weight medium, H), followed by CTBP2 co-immunoprecipitation (Fig 5A). Interestingly, ZNF217,

RCOR1, LSD1/KDM1A, and HDAC1/2, which are well-characterized CTBP1/2 interactors, more efficiently co-precipitated with CTBP2 when ZBTB18 was expressed (H/L = 1.83, 1.73, 1.6, and 1.48, respectively) (Fig 5B). CTBP2 co-immunoprecipitation in BTSC168 and SNB19 cells followed by Western blot analysis confirmed the increased binding of ZNF217, LSD1, and CTBP to each other, upon ZBTB18 expression (Fig 5C and D). Of note, in BTSC168 cells, LSD1 protein levels appeared to be higher upon ZBTB18 expression, which suggests that the protein could be stabilized (Fig 5D). Overall, these pieces of evidence indicate that CTBP2 complex dynamics and functions could change upon ZBTB18 overexpression. To verify whether ZBTB18 binding to CTBP2 affects the recruitment and/or stabilization of the CTBP2 complex at *SREBP* gene promoters, ChIP experiments were performed in SNB19 cells upon overexpression of ZBTB18, using LSD1- and ZNF217-directed antibodies. Here, both LSD1 and ZNF217 appeared to be more enriched when ZBTB18 was expressed (Fig 5E and F). We then used primary GBM cell BTSC168 transduced with ZBTB18 to validate the previous results by examining two representative *SREBP* gene promoters (*SCD* and *FASN*). The results confirmed that ZBTB18 binding to the tested promoters is accompanied by increased enrichment of CTBP2, LSD1, and ZNF217 in the same promoter region (in proximity of the transcription start) (Fig S10A). Interestingly, the expression of ZBTB18-mut, which cannot bind CTBP, did not produce the same effect (Fig S10A), suggesting that ZBTB18 interaction with CTBP is required to affect the composition/stability of those factors at the SREBP gene promoters. Because LSD1 is mostly implicated in gene repression through demethylation of H3K4me2, we asked whether this activity is affected by ZBTB18. Measurement of H3K4me2 methylation from SNB19 total lysate using a commercial assay showed that ZBTB18, but not ZBTB18-mut, impairs LSD1 activity (Fig 5G). A similar effect was caused by treatment with RN1, an irreversible LSD1 inhibitor (Fig 5G). As such, our data suggest that ZBTB18 promotes the interaction/stabilization of CTBP, LSD1, and ZNF217 at the *SREBP* gene promoters with a concomitant reduction in LSD1 demethylase activity. In line with this idea, the previously observed increase in H3K9me2 at the *SREBP* gene promoters upon ZBTB18 expression (Fig S9B) could be the result of a reduced LSD1 demethylase activity, because an LSD1-dependent H3K9me2 demethylase activity has also been described (Metzger et al, 2005). Consistent with these data, the ectopic expression of ZBTB18 in BTSC168 caused enrichment of H3K4me2 and H3K9me2 marks at selected *SREBP* gene promoters (Figs 5H and S10B). Interestingly, exposure to RN1 did not produce an additional effect in ZBTB18-expressing cells, where the LSD1 demethylase activity was already impaired by ZBTB18 (Figs 5H and S10B). Overall, the observation that ZBTB18 expression augments H3K4me2 and H3K9me2 levels is consistent with the hypothesis that ZBTB18 inhibits LSD1-activating function. ChIP analysis of H3K4me2 and H3K9me2 enrichment at the *SREBP* gene promoter upon *ZBTB18* KO in BTSC475 cells showed that both

microscope. For each image, the average red channel intensity was measured with Photoshop, excluding black areas with no cells, when necessary. Error bars ± s.d. *$P$ < 0.05, **$P$ < 0.01, and ***$P$ < 0.001. **(G)** Bodipy TMR-X lipid staining of BTSC475 cells upon ZBTB18 KO (sgZBTB18#3 and sgZBTB18#4). Nuclei were counterstained with DAPI. Scale bar: 100 $\mu m$. **(G, H)** Quantification of Bodipy TMR-X lipid staining shown in (G). **(I)** Analysis of the $^{13}$C-labeled glucose incorporation into fatty acids in BTSC168 transduced with empty vector (EV) or ZBTB18-overexpressing vector (ZBTB18). The four principal lipid species emerged from the analysis are individually represented to show the differential progressive incorporation of $^{13}$C. n = 3 biological replicates; error bars ± s.d. *$P$ < 0.05, **$P$ < 0.01, and ***$P$ < 0.001 by a multiple unpaired *t* test.

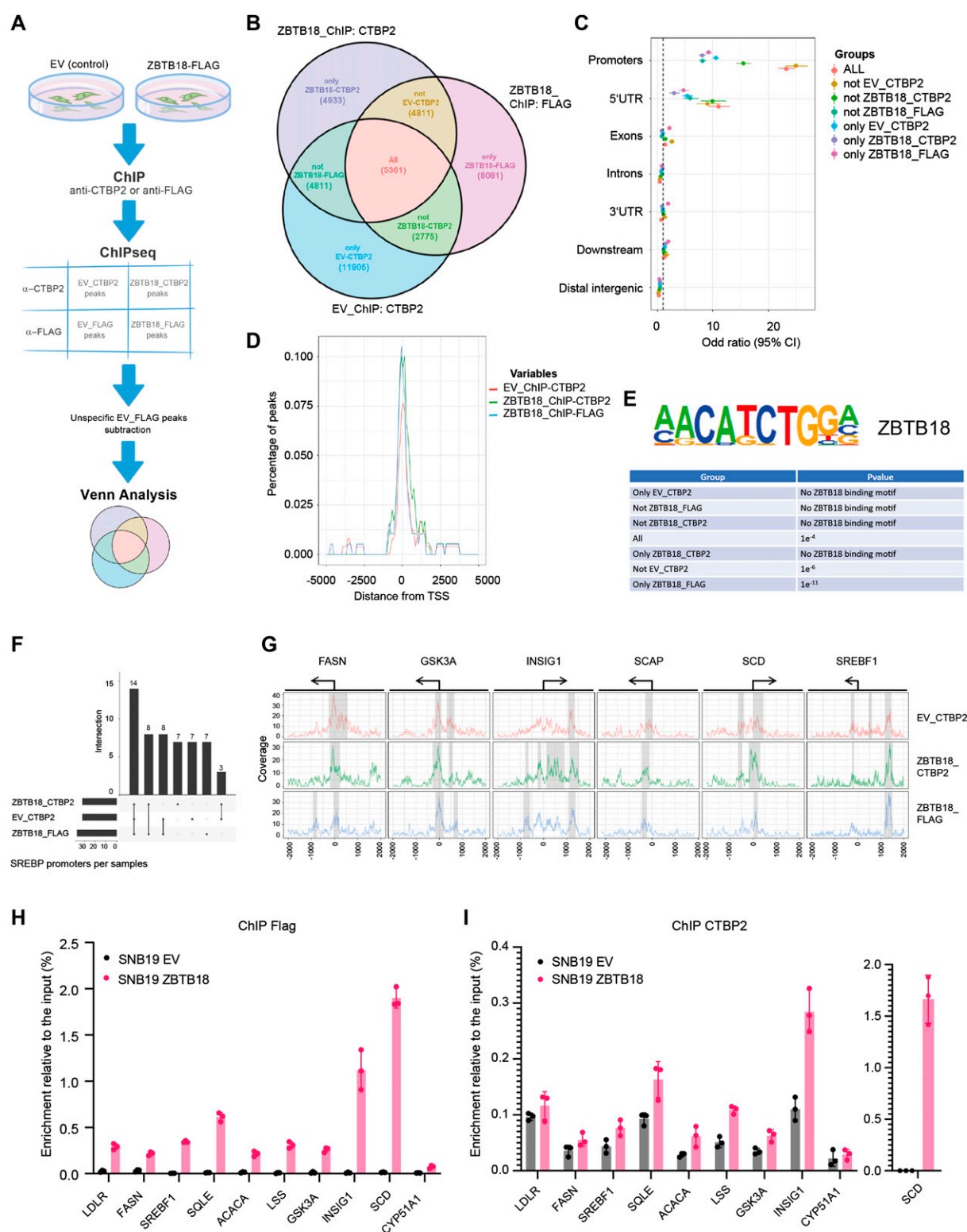

**Figure 4. CTBP2 and ZBTB18 map to the promoter of sterol regulatory element–binding protein (SREBP) genes.**
**(A)** Experimental flowchart of the ChIP-seq analysis. **(B)** Venn diagram showing the overlap between the three consensus peak sets from EV_CTBP2, ZBTB18_CTBP2, and ZBTB18_FLAG. **(C)** Enrichment of peaks on promoter regions depicted on a forest plot from Fisher's test analysis. Dots indicate the odds ratio, and lines indicate the 95% confidence interval. **(D)** Meta-gene peak density profile showing an overrepresentation of peaks around SREBP gene TSS regions. Y-axis indicates the percentage of peaks in EV_CTBP2, ZBTB18_CTBP2, and ZBTB18_FLAG that overlap the genomic region depicted on the x-axis. **(E)** Homer analysis showing enrichment of the ZNF238/ZBTB18 DNA binding motif in the Venn diagram groups. **(F)** UpSet plot showing the overlap of SREBP genes having at least one peak in one of the three conditions: EV_CTBP2,

markers are reduced at the *SCD* and *LDLR* promoters (Fig 5I and J), which appeared to be slightly up-regulated upon *ZBTB18* KO (Fig S6A and C). Together, these data suggest that, through binding to CTBP, ZBTB18 favors the recruitment/stabilization of LSD1 and ZNF217 to the SREBP gene promoters. At the same time, LSD1 demethylase activity appears to be impaired.

### ZBTB18 inhibits LSD1 demethylation activity to promote its scaffolding function

To better understand LSD1's role in the proposed scenario, we analyzed SREBP gene expression in glioblastoma cells, GBM#22 cells, in which LSD1 has been knocked out by CRISPR/Cas9, as recently described (Faletti et al, 2021). Interestingly, most of the SREBP genes tested were down-regulated upon *LSD1* knockdown (Fig 6A and B), suggesting that, similar to CTBP, LSD1 acts as a transcriptional activator of the SREBP genes. We performed similar experiments using a short hairpin RNA directed to LSD1 in GBM#22 and BTSC475 (Fig S11A–D); here, SREBF1 and SCD were confirmed to be the major LSD1 targets. ChIP analysis of H3K4me2 and H3K9me2 marks at the promoter of SREBP genes upon LSD1 knockdown showed enrichment for both histone marks at the *SREBF1* locus, whereas for the other genes tested, H3K9me2 appeared to be particularly enriched (Fig 6C). We then examined the expression levels of SREBP genes upon ZBTB18 expression and *LSD1* KO in GBM#22. Interestingly, *LSD1* depletion did not rescue ZBTB18-mediated SREBP gene repression (Fig 6D), suggesting that LSD1 lies downstream of ZBTB18, similar to what we had observed upon CTBP2 knockdown (Fig 2E). This further reinforces the hypothesis that ZBTB18 interaction with CTBP is not required for ZBTB18-mediated gene repression but rather inhibits CTBP- and LSD1-mediated activation. We then tested the effect of LSD1 knockdown and ZBTB18 expression in fatty acid synthesis by performing a $^{13}$C-labeled glucose tracing experiment. Consistent with the effect on SREBP gene expression, LSD1 depletion decreased glucose-derived $^{13}$C incorporation (Fig 6E). The same effect was observed in the presence of ZBTB18 (Fig 6E), further corroborating our hypothesis that LSD1 induces fatty acid synthesis and that ZBTB18 acts by repressing LSD1-mediated function.

Our data so far indicate that ZBTB18 inhibits LSD1 activity while concomitantly favoring its interaction with other protein partners such as ZNF217. To confirm whether inhibition of LSD1 demethylase activity is important for LSD1 interaction with its complex, we treated BTSC168 and BTSC268 cells with RN1. Interestingly, LSD1, ZNF217, and CTBP interaction was stabilized in the presence of RN1 (Fig 6F and G), further indicating that inhibition of LSD1 demethylase activity favors its scaffolding function and the interaction with other co-factors (i.e., ZNF217). In conclusion, we propose that ZBTB18 represses SREBP gene expression by inhibiting CTBP- and LSD1-mediated transcriptional activation (Fig 7A and B). Moreover,

ZBTB18 could promote LSD1 scaffolding function favoring the interaction between LSD1, CTBP, and ZNF217 (Fig 7C). In the future, more studies will be required to investigate whether the scaffolding role of LSD1 and its complex components (i.e., ZNF217) contributes to ZBTB18-mediated SREBP gene repression. Deciphering the precise molecular mechanism of fatty acid synthesis regulation will help define new potential therapeutic targets in GBM.

## Discussion

Here, we have identified a new role of ZBTB18, CTBP, and LSD1 in the regulation of fatty acid synthesis, which is considered a hallmark of cancer, including GBM. We show that the tumor suppressor ZBTB18 interacts with the co-factors CTBP1/2 and represses the expression of SREBP genes, involved in de novo lipogenesis. CTBP is known to be associated with protein complexes containing the histone demethylase LSD1. We show that in GBM cells, which express no or low levels of ZBTB18, CTBP and LSD1 activate the expression of SREBP genes; however, when ectopically expressed, ZBTB18 leads to SREBP gene repression by inhibiting CTBP-associated complex activity. Consistent with such a new function, ZBTB18 expression is paired with the reduction in several phospholipid species, which rely on fatty acid availability to be assembled, and with decreased lipid droplet content inside the cells (Fig 7). Interestingly, SREBPs have been recently implicated in mesenchymal transition, in glioblastoma, breast cancer, and endothelial cells (Zhang et al, 2019; Martin et al, 2021; Schmitt et al, 2021). Therefore, repressing SREBP-regulated genes could be an additional mechanism through which ZBTB18 counteracts mesenchymal transformation in non-mesenchymal gliomas, which others and we previously reported (Fedele et al, 2017).

### CTBPs act as transcriptional activators of SREBP genes

The interaction between ZBTB18 and CTBP2 initially prompted us to consider CTBP2 as a putative ZBTB18 co-repressor. However, genome-wide gene expression analysis indicated that ZBTB18 and CTBP2 play opposite roles in the transcriptional regulation of a common set of genes. Among them, EMT and SREBP genes are highly represented and appear to be activated by CTBP2 and repressed by ZBTB18. Many studies focusing on cell differentiation have shown that CTBP2 can prime its position in actively transcribed genes and that its binding to different transcription factors can cause the switch to a repressive state (Boxer et al, 2014; Kim et al, 2015). Conversely, it has been shown that CTBP can also be part of activator complexes; for example, CTBP2 and its associated proteins, LSD1 and NCOR1, have been implicated in transcriptional activation by the basic helix–loop–helix transcription factor NeuroD1, in gastrointestinal endocrine cells (Ray et al, 2014). Moreover,

---

ZBTB18_CTBP2, and ZBTB18_FLAG. **(G)** Read coverage around biologically relevant selected genes, that is, FASN, GSK3A, INSIG1, SCAP, and SCD. Identified peaks are highlighted under the grey area, and gene TSS regions are drawn on the top of the plot. **(H, I)** FLAG-ZBTB18 (H) and CTBP2 (I) enrichment at the promoter of the indicated SREBP genes in SNB19 cells transduced with empty vector (EV) of FLAG-ZBTB18. ChIP was performed using control beads alone (IgG), anti-FLAG antibodies, or anti-CTBP2 antibodies. Graphs show representative qRT–PCR results (n = 3 technical replicates) of at least three (FLAG) or two (CTBP2) biological replicates, which are expressed in % of the input as indicated.

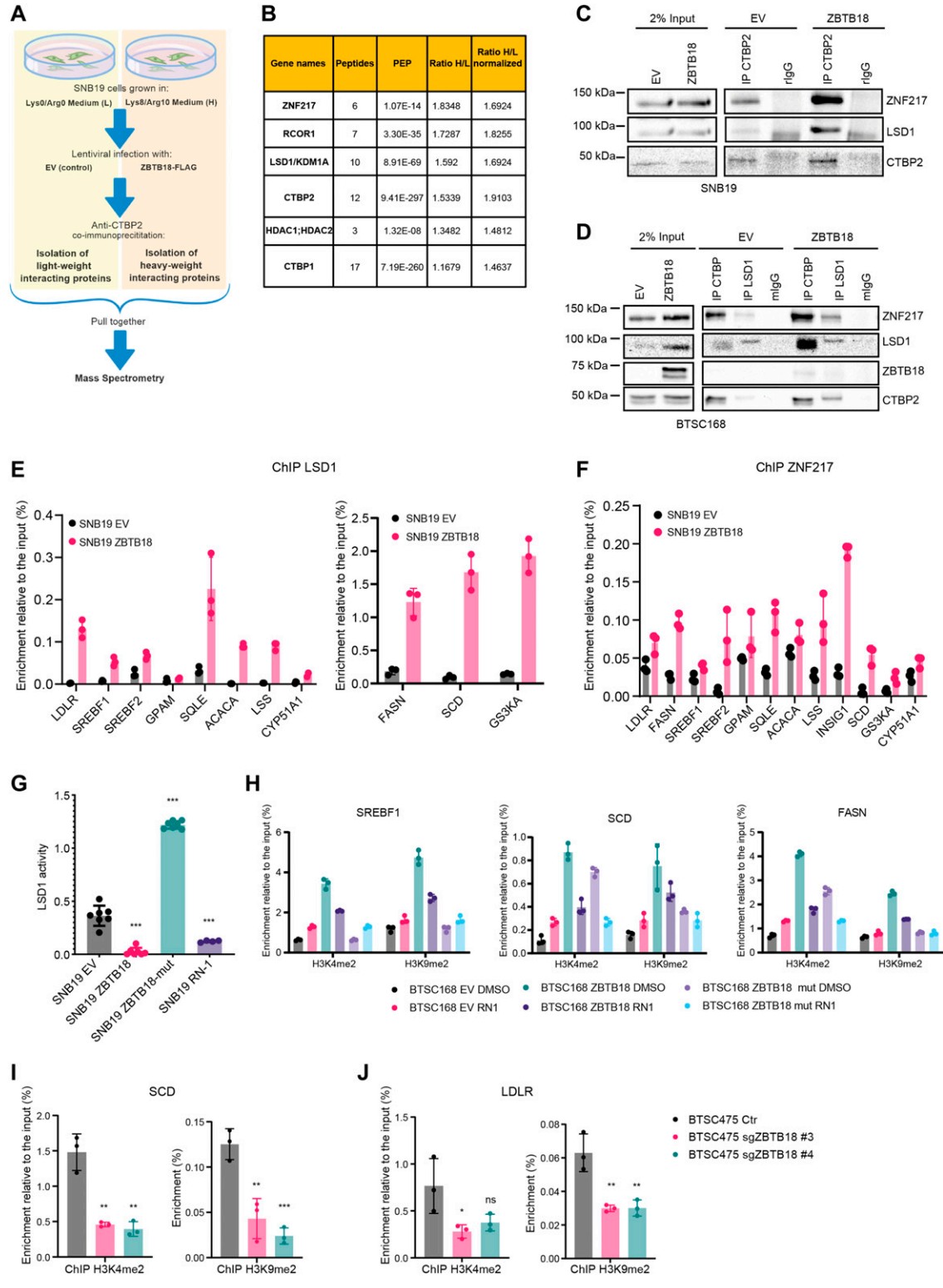

**Figure 5. ZBTB18 affects the CTBP2 complex activity at the sterol regulatory element–binding protein gene promoters.**
**(A)** Experimental flowchart of the SILAC-based MS analysis. **(B)** List of known CTBP2-interacting proteins identified by SILAC. Binding to CTBP2 increases in the presence of ZBTB18 (H) compared with EV (L). **(C)** WB analysis of CTBP co-IP in SNB19 cells transduced with EV or FLAG-ZBTB18. **(D)** WB analysis of CTBP and LSD1 co-IP in BTSC168 cells transduced with EV or FLAG-ZBTB18. **(E, F)** qRT-PCR showing binding of LSD1 (E) and ZNF217 (F) at selected sterol regulatory element–binding protein gene promoters in SNB19 cells transduced with EV, ZBTB18, and ZBTB18-mut. ZNF217 and LSD1 binding increases in the presence of ZBTB18. Graphs show representative qRT-PCR results (n = 3 technical replicates) of at least two biological replicates, which are expressed in % of the input as indicated. **(G)** Bar chart showing LSD1 activity in SNB19 cells transduced with EV, ZBTB18, and ZBTB18-mut (all n = 7), or treated with the LSD1 inhibitor RN-1 (n = 4). LSD1 activity is reduced upon ZBTB18 expression. **(H)** Representative

CTBP2 has been reported to act as a co-activator of the retinoic acid receptor (RAR) (Bajpe et al, 2013).

Recently, CTBP2 has been linked to the inhibition of cholesterol synthesis in breast cancer cells through direct repression of SREBF2 expression (Zhao et al, 2019). Furthermore, CTBP2 was shown to repress fatty acid biosynthesis in the liver (Sekiya et al, 2021). Although apparently in contrast to our proposed role of CTBP2 as an activator of SREBP genes, it is possible that CTBP function might change based on the interaction with other transcription factors. According to this possibility, CTBP2 repression of fatty acid in breast cancer requires the interaction with the transcriptional repressor ZEB1, which was never detected as a CTBP2 partner in our co-immunoprecipitation analysis in GBM cells. Instead, the study performed by Sekiya et al (2021) proposes that CTBP2 interacts with the SREBF1 protein to directly inhibit its transcriptional activity. However, these proposed models are mechanistically different from the one reported in our study, in which the newly described CTBP function seems to depend on the activating role of LSD1. This new role is consistent with previous studies, showing that CTBP and LSD1 can be implicated in transcriptional activation. It would be interesting to investigate whether other transcription factors participate in CTBP- and LSD1-mediated SREBP gene activation.

### LSD1 plays a role as an activator of SREBP genes

Our further analysis suggests that ZBTB18 could interfere with CTBP transcriptional activation of SREBP genes by inhibiting the enzymatic activity of its associated complex, including LSD1-mediated deme-thylation of H3K4me2 and H3K9me2. Although most of the studies indicate that LSD1 demethylates mono- and di-methylated H3K4, its role as H3K9me2 demethylase has also been reported, mostly in association with nuclear receptors (Metzger et al, 2005; Garcia-Bassets et al, 2007). Although it cannot be excluded that increased levels of H3K9me2 upon ZBTB18 expression could result from other co-repressor activities (i.e., specific histone methyltransferases), our data using both LSD1 knockdown and the inhibitor RN1 indicate that, in GBM cells, LSD1 also displays an H3K9me2 demethylase activity, which is important for the LSD1-mediated expression of SCD and SREBF1. Interestingly, both positive and negative changes of H3K9me2 have been implicated in the development and progression of various types of cancer because of its influence on cell differentiation, apoptosis, and treatment response (Schulte et al, 2009; Chen et al, 2015). Thus, the role of H3K9me2 in promoting or restringing cancer might depend on the genomic context and on the activity of specific histone methylation and demethylation enzymes. The observation that LSD1 knockout and inactivation cause an increase in both dimethyl H3K4 and H3K9, especially at the *SREBF1* locus, is in line with previous findings by Garcia-Bassets and colleagues, who demonstrated that both H3K4me2 and H3K9me2 marks are simultaneously decreased upon ER- and LSD1-dependent gene activation (Garcia-Bassets et al,

2007). Interestingly, the author also showed that LSD1 contributes to other TF-mediated activation, namely, NF-κB, AP-1, and βRAR. Furthermore, our data indicate that, in the presence of ZBTB18, LSD1 and CTBP more efficiently interact with each other and with the transcriptional co-repressor ZNF217. Similar results were obtained when LSD1 demethylase activity was inhibited by RN1. Therefore, we speculate that blocking LSD1 demethylase activity could favor LSD1 stability and scaffolding function. This hypothesis is consistent with recent studies, indicating that LSD1 can repress gene expression independently from its histone demethylase activity (Sehrawat et al, 2018; Ravasio et al, 2020). In particular, Sehrawat and colleagues demonstrated that LSD1 gene regulation in prostate cancer is mediated by interaction with ZNF217 independently of its demethylase activity. However, whether the scaffolding function of LSD1 and the recruitment of co-repressors such as ZNF217 further contribute to ZBTB18-mediated SREBP gene repression needs further studies to be elucidated.

### ZBTB18 expression affects the content of lipids in the cell

Our lipidomics study shows that the expression of ZBTB18 in GBM cells affects the synthesis of phospholipids. An elevated phospholipid production rate is a trait of rapidly dividing cells, such as tumor cells, as they are required to form the cellular membranes of the new cells. The reduced amount of several phospholipid species upon ZBTB18 ectopic expression might be linked to the previously observed decreased proliferation rate (Fedele et al, 2017).

The abundance of lipid droplets within the cell depends on several factors, including de novo lipid synthesis, uptake of lipids from the environment, and mobilization rate, according to the lipid demand for cellular functions (Olzmann & Carvalho, 2019). Our data suggest that ZBTB18 specifically affects the lipid biosynthesis but not the ability of GBM cells to gather lipids from external sources. In fact, although most of the cells rely on external sources of fatty acids, cancer cells have been shown to be able to synthesize their own lipids and recent studies have provided evidence of a relevant role of fatty acid oxidation on GBM tumor growth (Duman et al, 2019). The negative effect of ZBTB18 expression on lipogenesis is further supported by the reduced incorporation of glucose-derived $^{13}$C isotopes in newly synthesized fatty acids. Even though this assay is biased by the presence in the culture medium of other potential carbon atom sources (i.e., glutamine) and of lipids that can be directly taken up by the tumor cells, the results added up to the rest of the data, pointing to a role of the ZBTB18 complex in hampering de novo lipogenesis.

When we attempted to knock down *ZBTB18* in BTSC475, a primary GBM cell line that shows some basal level of ZBTB18, we observed increased lipid droplet accumulation upon ZBTB18 loss. However, the level of SREBP gene re-expression is modest, suggesting that other mechanisms, which affect lipid droplet turnover, could be involved. It is worth mentioning that because ZBTB18 is almost

ChIP-qRT-PCR showing enrichment of H3K4m2 and H3K9me2 marks in BTSC168, transduced with empty vector (EV), FLAG-ZBTB18, or FLAG-ZBTB18mut, and treated with the LSD1 inhibitor RN1, at the *SREBF1*, *SCD*, and *FASN* promoters. Graphs show representative qRT-PCR results (n = 3 technical replicates) of at least two biological replicates, which are expressed in % of the input. **(I, J)** qRT-PCR showing binding of H3K4me2 and H3K9me2 at the *SCD* (I) and *LDLR* (J) gene promoters in BTSC475 cells transduced with sgZBTB18 #3 and #4. Graphs show the average of three independent qRT-PCR results that are expressed in % of the input as indicated. Error bars ± s.d. *P < 0.05, **P < 0.01, and ***P < 0.001 by a *t* test.

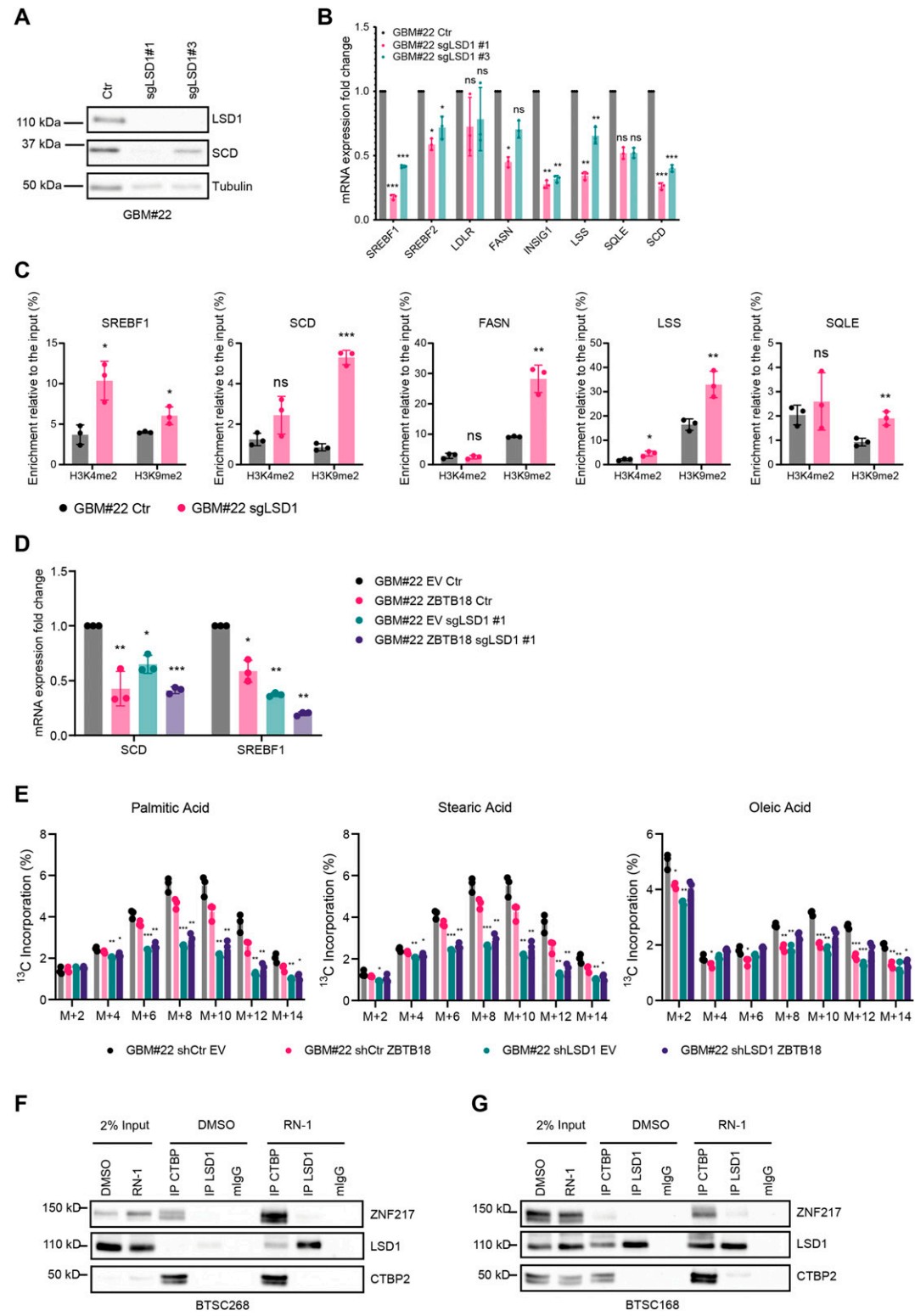

**Figure 6. LSD1 functions as an activator of sterol regulatory element–binding protein (SREBP) genes and requires H3K4 and H3K9 demethylase activity.**
**(A)** WB analysis of LSD1 and SCD expression in GBM#22 cells upon CRISPR/Cas9-mediated *LSD1* knockdown with two sgRNAs. **(B)** qRT-PCR showing the expression levels of selected SREBP genes upon LSD1 depletion. n = 3 biological replicates; error bars ± s.d. *$P < 0.05$, **$P < 0.01$, and ***$P < 0.001$ by a $t$ test. Gene expression was normalized to 18S RNA. **(C)** qRT-PCR showing enrichment of H3K4me2 and H3K9me2 marks at the promoter of the indicated *SREBP* genes, upon *LSD1* knockout in GBM#22 cells. n = 3 biological replicates; error bars ± s.d. *$P < 0.05$, **$P < 0.01$, and ***$P < 0.001$ by a $t$ test. Results are expressed in % of the input as indicated. **(D)** qRT-PCR showing expression levels of SREBF1 and SCD genes upon *LSD1* knockout in the presence or absence of ectopic ZBTB18. n = 3 biological replicates; error bars ± s.d. *$P < 0.05$, **$P <$

uniformly low or absent in GBM cells, knockout studies should be performed using low-grade glioma cells, in which ZBTB18 is more expressed. However, the proper establishment of low-grade glioma-derived cells has so far represented a major obstacle for this kind of study. Of note, patient-derived organoid models of lower grade gliomas have been recently established (Abdullah et al, 2022), which could represent a useful model for future investigation of the ZBTB18 function. In conclusion, we unravel a new epigenetic mechanism of transcriptional regulation employed by the tumor suppressor ZBTB18 to repress SREBP genes and de novo lipogenesis in GBM. Epigenetic changes have emerged to be a critical step for tumorigenesis and metastasis, but the key events in cancer cell transformation still remain poorly understood. Our findings contribute to overcoming this gap of knowledge. Because of the role of ZBTB18 as a negative regulator of the mesenchymal transformation in GBM, our results have important implications in cancer therapy as they might help to find more effective strategies to diagnose and treat glioblastoma. Furthermore, given the recognized oncogenic role of CTBP and LSD1 in other cancers, their implication in de novo lipogenesis could have a broader impact on cancer research.

# Materials and Methods

## Cell culture

SNB19, LN229, and HEK293T cells were cultured as previously described (Fedele et al, 2017). For lipid starvation, SNB19 cells were grown in DMEM supplemented with 10% lipid-depleted FBS (Biowest). Primary glioblastoma stem cells such as BTSC233, BTSC268, BTSC168, and BTSC475 were generated in our laboratory in accordance with an Institutional Review Board–approved protocol (Fedele et al, 2017). U3082 were generated at the University of Uppsala (Xie et al, 2015) and kindly provided by Dr. Nelander. Primary glioblastoma stem cells such as GBM#22 were established at the European Institute of Oncology and kindly gifted by Dr. Pelicci. All primary glioblastoma stem cells were grown in a Neurobasal medium (Life Technologies) containing B27 supplement (Life Technologies), FGF (20 ng/ml, R&D Systems), EGF (20 ng/ml, R&D Systems), (LIF 20 ng/ml, Genaxxon Biosciences), heparin (2 g/ml; Sigma-Aldrich), and GlutaMAX (Invitrogen). U3082 were grown adherent on dishes previously coated with laminin (Life Technologies). All cells were mycoplasma-free. SNB19 LN229 and HEK293T cells have been authenticated on 3/2/2017 (SNB19 and LN229) and on 11/26/19 (SNB19 and HEK293T) by PCR–single-locus technology (Eurofins Medigenomix). For CTBP1/2 inhibition, SNB19 and BTSC233 cells were treated with 10 mM 4-methylthio-2-oxobutyric acid (MTOB; Sigma-Aldrich) for 24 h. To inhibit LSD1, GBM cells were treated with 5 μM of RN1 (Sigma-Aldrich) for 96 h.

## RNA extraction and quantitative real-time PCR

Total RNA was extracted from the cell culture using miRNeasy Mini Kit (QIAGEN) according to the manufacturer's instructions. First-strand cDNA synthesis was generated using the Superscript III First-Strand Synthesis System for RT–PCR (Life Technologies) following the manufacturer's protocol. Quantitative RT–PCR was performed using Kapa SYBR Fast (Sigma-Aldrich). SREBP gene primer sequences are listed in Table S1. Primers for SERPINE1, TNFAIP6, CD97, LGALS1, S100A6, and ID1 were previously described (Fedele et al, 2017).

## Gene expression analysis

For gene expression analysis, 1.5 μg of total RNA was processed and analyzed at the DKFZ. Hybridization was carried out on Illumina HumanHT-12v4 Expression BeadChip. Microarray data were further analyzed by GSEA (http://www.broadinstitute.org/gsea/index.jsp). The microarray gene accession number is GSE138890. Differentially regulated genes were identified using the limma R package (Ritchie et al, 2015). Genes with adjusted *P* < 0.05 and absolute fold change > 0.5 were considered as significantly regulated. A gene set enrichment analysis was performed using Fisher's exact test on the ConsensusPathDB (Kamburov et al, 2013), comparing these selected genes against the whole set of quantified genes. The significance threshold was set to adjusted *P* < 0.05. Gene correlation, SREBF1 gene expression, and patient survival analyses were performed using the GlioVis platform (Bowman et al, 2017) and data from The Cancer Genome Atlas (TCGA) (https://www.cancer.gov/tcga) and the Chinese Glioma Genome Atlas (CGGA) Project (Zhao et al, 2017).

## Immunoblotting

Total protein extracts were prepared as previously described (Fedele et al, 2017). Western blots were performed using the antibodies listed in Table S2.

## Lentiviral vectors

To produce FLAG-ZBTB18-HisZBTB18, ZBTB18 was PCR-amplified from the previously described pCHWMS-eGFP-ZBTB18 lentiviral vector (Fedele et al, 2017) using primers containing BstXI and PmeI restriction sites. Upon restriction digestion, the ZBTB18 fragment was cloned into BstXI and PmeI sites by removing the LUC region of the PCHMWS-eGFP-IRES vector. Primers are listed in Table S3. ZBTB18 LDL-mut was obtained by site-directed mutagenesis using pCHWMS-eGFP-FLAG-ZBTB18-His as a template and the QuikChange II XL Site-Directed Mutagenesis kit (Agilent), according to the manufacturer's instructions. All constructs were sequence-validated.

Lentiviral stock production and cell infection were performed as previously described (Fedele et al, 2017). For CTBP1 and CTBP2 knockdown, the following short hairpin RNAs cloned in the pLKO

---

0.01, and ***P < 0.001 by a t test. **(E)** Analysis of the $^{13}$C-labeled glucose incorporation into fatty acids in GBM#22 cells upon CRISPR/Cas9-mediated *LSD1* knockdown (sgLSD1#1) and transduced with empty vector (EV) or ZBTB18-overexpressing vector (ZBTB18). The three principal lipid species emerged from the analysis are individually represented to show the differential progressive incorporation of $^{13}$C. n = 3 biological replicates; error bars ± s.d. *P < 0.05, **P < 0.01, and ***P < 0.001 by a multiple unpaired *t* test. **(F, G)** WB analysis of LSD1 or CTBP co-IP in BTSC268 (F) and BTSC168 (G) cells upon treatment with RN-1.

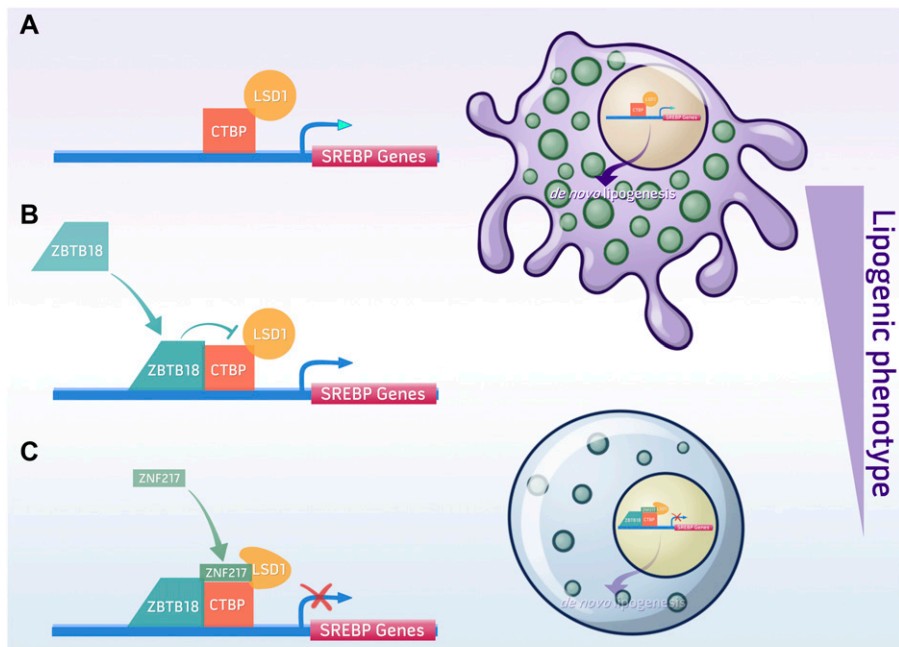

**Figure 7. Model of ZBTB18- and CTBP2-mediated regulation of sterol regulatory element–binding protein (SREBP) gene expression.**
**(A)** In the absence of ZBTB18, CTBP and LSD1 activate the expression of SREBP genes. Other TFs or co-factors could be involved. **(B, C)** When expressed, ZBTB18 inhibits LSD1 demethylase activity (B) and recruits a repressive complex at the promoter of SREBP genes (C).

lentiviral vector (Sigma-Aldrich) were used: shCTBP2-#1(TRCN0000013745), shCTBP2#2 (#TRCN0000013747), shCTBP1#8 (TRCN0000013739), and shCTBP1#9 (TRCN0000273844), or a non-targeting shRNA (SHC002, here shCtr). LSD1 silencing was achieved by lentiviral transfection with MISSION pLKO.1-puro Empty Vector Plasmid DNA (Sigma-Aldrich) harboring either the sequence targeting human LSD1 (TRCN0000046071) or a non-targeting shRNA (SHC002, here shCtr).

For *ZBTB18* KO, four sgRNAs targeting exon 2 (*ZBTB18* KO#1, AAAGTCGAGAGTCTCTCCGA; *ZBTB18* KO#2, ATCTGCCGAATCCCTCACGG; *ZBTB18* KO#3, CAAGCAGGAGAGCGAAAGCG; and *ZBTB18* KO#4, AAGTGTGAGCACTAATAACA) were cloned in the pKLV-U6gRNA(BbsI)-PGKpuro2BFP (#50946; Addgene) and lentiviral particles were prepared as described above to transduce BTSC475 cells. Cells were selected with 1 μg/ml puromycin, transduced with Cas9-expressing lentiviral particle (pLentiCas9-GFP, #86145; Addgene), and selected with 5 μg/ml blasticidin. *ZBTB18* knockout was verified by Sanger sequencing and Western blot.

*LSD1* KO in GBM#22 were generated by the Cogentech Genome Editing Facility and kindly provided by Dr. Pelicci. Two different sgRNA target sequences directed to three different exons were used: *LSD1* KO#1, GGTAATTATTATAGGCTCTG (E7T13) (exon 7); and *LSD1* KO#3, CTAAATAACTGTGAACTCGG (E6T1) (exon6). Guide RNAs were cloned in an all-in-one PX458 lentiviral vector using Nucleofection (Kit V, Program T-020). GFP-positive cells were sorted after 48 h, and single clones were established by limiting dilution. After single-clone propagation, *LSD1* knockout was verified by NGS (Ion Proton) and Western blot.

## Co-immunoprecipitation, mass spectrometry, and SILAC

For co-immunoprecipitation (co-IP), GBM cells were lysed in 1 ml of co-IP buffer (150 mM NaCl, 50 mM Tris–HCl, pH 7.5, 10% glycerol, 0.2% Igepal, and 1 mM EDTA) supplemented with Halt protease and phosphatase inhibitor cocktail (1 mM; Thermo Fisher Scientific), and PMSF (1 mM; Sigma-Aldrich), vortexed for 30 s, and kept on ice for 30 min. After centrifugation at 16,000*g* for 20 min at 4°C, a pre-clearing step was performed by adding 25 μl of Protein A/G Plus Agarose beads (Santa Cruz) and incubating the samples for 1 h at 4°C in rotation. The recovered supernatant was incubated over-night with the antibodies listed in Table S4. 20 μl of lysate (input) was removed and mixed with an equal volume of 2× Laemmli buffer for Western blot analysis. 20 μl of equilibrated protein A/G beads was added to each lysate and incubated for 2 h at 4°C. The beads were then washed four times with 1 ml of co-IP buffer before eluting the co-precipitated proteins with 20 μl of 2× Laemmli buffer. Samples were stored at –20°C or directly analyzed by Western blot. co-IP and subsequent MS analyses in U3082 have been previously described (Masilamani et al, 2022).

To run SILAC, SNB19 cells were SILAC-labeled using Arg0/Lys0 (low, L) and Arg10/Lys8 (high, H) (Silantes). Afterward, cells were transduced with pCHMWS-EV (L) or pCHMWS-FLAG-ZBTB18 (H) lentiviral particles. Transduced SNB19 were mixed and lysed as described above. Total protein extracts were incubated with rabbit anti-CTBP2 (#13256S; Cell Signaling), and the precipitated fraction was subjected to MS to identify interacting proteins and calculate H/L ratios. For MS analysis, 1 μg of peptides was analyzed on a Q Exactive Plus mass spectrometer (Thermo Fisher Scientific) coupled to an EASY-nLCTM 1000 UHPLC system (Thermo Fisher Scientific). The analytical column was self-packed with silica beads coated with C18 (ReproSil-Pur C18-AQ, d = 3 Å) (Dr. Maisch HPLC GmbH). For peptide separation, a linear gradient of increasing buffer B (0.1% formic acid in 80% acetonitrile, Fluka) was applied, ranging from 5 to 40% buffer B over the first 90 min and from 40 to 100% buffer B in the subsequent 30 min (120 min separating gradient length). Peptides were analyzed in a data-dependent acquisition (DDA)

mode. Survey scans were performed at 70,000 resolution, an AGC target of $3 \times 10^6$, and a maximum injection time of 50 ms followed by targeting the top 10 precursor ions for fragmentation scans at 17,500 resolution with 1.6 m/z isolation windows, an NCE of 30, and a dynamic exclusion time of 35 s. For all MS2 scans, the intensity threshold was set to $1.3 \times 10^5$, the AGC to $1 \times 10^4$, and the maximum injection time to 80 ms. Raw data were analyzed with MaxQuant (v 1.6.14.0) allowing two missed cleavage sites, no variable modifications, carbamidomethylation of cysteines as fixed modification, label-free quantification (LFQ), and match between runs (MBR) set to activated. The Human-EBI-reference database was downloaded from https://www.ebi.ac.uk/ on January 9, 2020. Only unique peptides were used for quantification. All measurements and analyses were performed at the Core Facility Proteomics of the Center for Biological System Analysis (ZBSA) at the University of Freiburg. MS analysis of ZBTB18 co-IP in BTSC268 was performed at the Proteomic Platform—Core Facility of the Freiburg Medical Center.

## Migration, proliferation, and apoptosis assays

Migration, proliferation, and apoptosis assays were performed as previously described (Fedele et al, 2017). For apoptosis assay, transduced SNB19 cells were seeded in a 96-well (dark) plate at a density of $1 \times 10^3$ per well. The next day, activity of caspases 3 and 7 was measured using the Caspase-Glo 3/7 Assay (Promega), according to the manufacturer's instructions.

## LSD1 activity assay

For the LSD1 activity assay, SNB19 cells were transduced with EV, ZBTB18, or ZBTB18-mut as described above or treated with 5 $\mu$M RN-1 inhibitor (Calbiochem), for 96 h. Nuclear lysates were prepared with the EpiQuik Nuclear Extraction Kit (EpiGentek). Measurement of LSD1 activity was performed with Epigenase LSD1 Demethylase Activity/Inhibition Assay Kit (EpiGentek) according to the manufacturer's instructions. LSD1 activity was quantified by Infinite 200 PRO Spectrophotometer (Tecan) at 450 and 655 nm, following the manufacturer's instructions.

## ChIP-seq and quantitative ChIP

For ChIP-seq, SNB19 cells were transduced and incubated with ChIP Crosslink Gold solution (Diagenode) and processed using the iDeal ChIP-seq kit for Transcription Factors (Diagenode), according to the manufacturer's instruction. Antibodies are listed in Table S5. Library preparation and ChIP-seq were performed at Diagenode (https://www.diagenode.com). ChIP-seq gene accession number is GSE140002. Downstream analyses were executed with R (3.6.0). For each condition, two independent ChIP experiments were performed. Venn analysis was performed with the ChIPpeakAnno R package (Zhu et al, 2010). A single-base overlap threshold was used to identify the common peaks between the three conditions. Peaks were divided according to the Venn subgroups and annotated with the nearest gene using ChIPseeker (Yu et al, 2015) and TxDb.Hsapiens UCSC.hg19.knownGene R packages. Enrichment analysis of gene regions was done via Fisher's exact test using 100,000 regions

of 200 bp, randomly selected on the human genome, as background. ReMap (Cheneby et al, 2020) was used to look at the overlap with other TF and co-factor peaks from published data in other cell lines. The statistical significance of the overlap was calculated by repeating the analysis using the same number of randomly selected regions of 2,000 bp for 1,000 times. The $P$ was calculated from the empirical cumulative distribution. Significance of the overlap was determined in the same way for SREBP gene TSS regions. Motif enrichment analysis was performed with the Homer software (Heinz et al, 2010).

Quantitative ChIP was performed as follows: SNB19 or BT168 cells were seeded at $1 \times 10^6$ cell confluence and transduced either with the control or with ZBTB18-expressing lentiviral vectors as described before. After 72 h, cells were washed in PBS and incubated in the presence of ChIP Crosslink Gold solution (Diagenode) for 30 min at room temperature, fixed by addition of 1% formaldehyde for 20 min at room temperature, and quenched with PBS for 5 min. The cells were resuspended in 2 ml L1 buffer (50 mM Tris, pH 8.0; 2 mM EDTA, pH 8.0; 0.1% NP-40; 10% glycerol; and protease inhibitors) per $10^7$ cells, and lysed on ice for 5–10 min. The nuclei were collected by centrifugation at 800$g$ for 5 min at 4°C and lysed in 1 ml of L2 buffer (0.2% SDS; 10 mM EDTA; 50 mM Tris, pH 8; and protease inhibitors). The suspension was sonicated in a cooled Bioruptor Pico (Diagenode) and cleared by centrifugation for 10 min at 15,700$g$. The chromatin (DNA) concentration was quantified using NanoDrop (Thermo Fisher Scientific), and the sonication efficiency was monitored on an agarose gel. Protein A Sepharose (PAS) beads (GE Healthcare) were blocked with sonicated salmon sperm DNA (200 mg/ml beads) and BSA (250 mg/ml beads) in dilution buffer (0.5% NP-40; 200 mM NaCl; 50 mM Tris, pH 8.0; and protease inhibitors) for 2 h at room temperature. The chromatin was diluted 10× in the dilution buffer and pre-cleared with blocked PAS for 1 h at 4°C. 5 $\mu$g of pre-cleared chromatin was incubated with 5 $\mu$g of antibody O/N at 4°C, then with 40 $\mu$l of blocked PAS for further 2 h at 4°C. Antibodies are listed in Table S5. Control mock immunoprecipitation was performed with blocked PAS. The beads were washed 4× with WB-250 (0.02% SDS; 0.5% NP-40; 2 mM EDTA; 250 mM NaCl; and 20 mM Tris, pH 8.0). The immunocomplexes were eluted by two 15-min incubations at 30°C with 100 $\mu$l elution buffer (1% SDS and 100 mM NaHCO$_3$) and de-crosslinked overnight at 65°C in the presence of 10U RNase (Roche). The immunoprecipitated DNA was then purified with the QIAquick PCR purification kit (QIAGEN) according to the manufacturer's protocol and analyzed by qRT–PCR. Primers are listed in Table S6.

## Lipidomics analysis

For lipid profiling, cells were washed, quenched, and lysed as previously described (Lagies et al, 2018) and lipids were extracted as described by Sapcariu et al (2014). The internal standard containing organic phase was evaporated to dryness and reconstituted in isopropanol:acetonitrile:water 2:1:1 and subjected to targeted LC/MS lipid profiling. A BEH C18 (2.1 × 100 mm, 1.8 $\mu$m) column (Waters Corporation) was used with the following chromatographic program: 40% solvent B (10 mM NH$_4$CHO$_2$, 0.1% formic acid, and isopropanol:acetonitrile 9:1) was held for 2 min, then increased to 98% B within 10 min (held for 2 min) and to 40% B within 0.5 min (held for

5.5 min). Solvent A was 10 mM $NH_4CHO_2$, 0.1% formic acid, and acetonitrile:water 3:2. The flow rate was set to 300 $\mu$l/minute, and the column temperature was 55°C. The method was further verified by analyzing lipid standards with the same chromatography coupled to a high-resolution mass spectrometer (Synapt G2Si; Waters Corporation). Samples were kept at 12°C and injected in a randomized order with pooled quality control samples injected at regular intervals. Lipids were monitored by MRM and SIM scan modes using an Agilent 6460 triple quadrupole. Raw data were analyzed by Agilent Quantitative Analysis software. Samples were normalized to an internal standard, QC-filtered, and range-scaled for principal component analysis and heatmap generation. Significance was determined by ANOVA including FDR multiple testing correction (q-value cutoff: 0.05). Statistical analyses and visualization were carried out by MetaboAnalyst 4.0 (Chong et al, 2018).

### $^{13}$C-labeled glucose incorporation analysis

Cells were grown as described above. The day before harvesting, cells were transferred into six-well plates at 400,000 cell/well density; each sample group was seeded in triplicate. The old culture medium was substituted by a new one with the same composition, but containing 10 mM $^{13}$C-labeled glucose (Cambridge Isotope Laboratories) instead of 10 mM glucose. After overnight incubation, cells were harvested and lysed as before. The $CHCl_3$ was evaporated, and the lipids were saponified by addition of 500 $\mu$l 1.5 M NaOH and incubated for 4 h at 80°C and 100$g$. Afterward, 500 $\mu$l 2 M HCl was added and fatty acids were extracted by adding 500 $\mu$l $CHCl_3$. After centrifugation (5 min, 15,000$g$, 4°C), 400 $\mu$l of the $CHCl_3$ phase was evaporated and derivatized for gas chromatography–electron ionization–mass spectrometry analysis by 20 $\mu$l pyridine and 50 $\mu$l N-methyl-N-trimethylsilyltrifluoroacetamide for 30 min at 37°C and 100$g$. Extracted ion chromatograms for the isotopologues for the different fatty acids (trimethylsilylated fatty acids –$CH_3$) were generated, integrated, and corrected for natural isotopic abundance. Relative abundance was calculated by dividing each isotopologue by the sum of the isotopologues of each fatty acid.

### Immunostaining and lipid staining

SNB19 cells were grown in four-well chamber slides either with 10% FBS/DMEM or with 10% lipid-depleted FBS (Biowest)/DMEM for 48 h. For lipid starvation, cells were kept for an additional 2 h either in 10% lipid-depleted FBS/DMEM or in 10% FBS/DMEM. All cells were then fixed with 4% paraformaldehyde in PBS and processed for the staining. ZBTB18 was labeled with anti-ZBTB18 rabbit polyclonal primary antibody (#12714-1-AP; Proteintech) and anti-rabbit IgG (H+L) Alexa Fluor 647 (goat; Thermo Fisher Scientific) secondary antibody. Lipid droplets were stained with 0.5 $\mu$g/ml Bodipy TMR-X SE (Thermo Fisher Scientific) in 150 mM NaCl for 10 min at room temperature. Nuclei were counterstained with 4′,6-diamidino-2-phenylindole (DAPI; Sigma-Aldrich). Pictures were acquired using a FSL confocal microscope (Olympus).

### Lipid uptake

SNB19 cells were seeded in four-well chamber slides and infected as described above. The cells were then incubated with 50 nM Bodipy-C16 (Thermo Fisher Scientific) in PBS for 15 min at room temperature. Subsequently, the samples were fixed with 4% paraformaldehyde in PBS, counterstained with DAPI, and imaged using a FSL confocal microscope (Olympus).

### Statistical analysis

For the statistical analysis, qRT–PCR data are judged to be statistically significant when $P < 0.05$ by a two-tailed $t$ test; $^{13}$C-labeled glucose incorporation analysis results were considered statistically significant when $P < 0.05$ by a multiple unpaired $t$ test. The number of replicates and the definition of biological versus technical replicates are indicated in each figure legend. All graphs and statistical analyses were generated using GraphPad Prism 9.

## Data Availability

The microarray data from this publication have been deposited to the GEO database (https://www.ncbi.nlm.nih.gov/geo/) and assigned the identifier GSE138890. For ChIP-seq data, the gene accession number is GSE140002. The protein interactions from this publication have been submitted to the IMEx consortium (www.imexconsortium.org) through IntAct (Orchard et al, 2014) and assigned the identification number IM-27496.

## Supplementary Information

## Acknowledgements

We thank Mahmoud Abdelkarim, Jonathan Goldner, Verena Haase, Marzieh Mohammadi, Sohie Gilles, Maria Denise Amico, and Jutith Treiber for technical assistance. We thank Veronica Dumit and Lisa Winner (University of Freiburg, ZBSA) for MS analysis and Marc Timmers for advice and discussion. We are grateful to M Squatrito (CNIO, Madrid, Spain) for sharing protocols and advice on CRISPR/Cas9 methodology. The results shown here are in part based upon data generated by TCGA Research Network: https://www.cancer.gov/tcga. This work was financed by the German's Cancer Aid grant (Deutsche Krebshilfe, 70113120) to MS Carro. M Boerries was supported by the Deutsche Forschungsgemeinschaft (DFG)—SFB1453 (Project ID 431984000-S1), SFB1160 (Project Z02), and TRR167 (Project Z01)—and by the German Federal Ministry of Education and Research by MIRACUM within the Medical Informatics Funding Scheme (FKZ 01ZZ1801B), and G Andrieux, by the Junior Research Group EkoEsted (FKZ 01ZZ2015). A Izzo was supported by the Research Committee (Forschungskommission) of the Faculty of Medicine, University of Freiburg. The Proteomic Platform—Core Facility was supported by the Research Committee (Forschungskommission) of the Faculty of Medicine, University of Freiburg.

## Author Contributions

R Ferrarese: formal analysis, investigation, visualization, methodology, and writing—original draft, review, and editing.
A Izzo: investigation, methodology, and writing—original draft.
G Andrieux: formal analysis and visualization.
S Lagies: formal analysis, visualization, and methodology.
JP Bartmuss: investigation.
AP Masilamani: investigation.
A Wasilenko: investigation.
D Osti: resources.
S Faletti: resources.
R Schulzki: investigation.
S Yuan: investigation.
E Kling: investigation.
V Ribecco: investigation.
DH Heiland: formal analysis.
S Tholen: formal analysis and methodology.
M Prinz: resources.
G Pelicci: resources.
B Kammerer: formal analysis and supervision.
M Borries: formal analysis, supervision, and funding acquisition.
MS Carro: conceptualization, supervision, funding acquisition, investigation, project administration, and writing—original draft, review, and editing.

## Conflict of Interest Statement

The authors declare that they have no conflict of interest.

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
