## [Reviewer comments · Life Science Alliance]

Life Science Alliance

ZBTB18 inhibits SREBP-dependent lipid synthesis by halting CTBPs and LSD1 activity in glioblastoma

Roberto Ferrarese, Annalisa Izzo, Geoffroy Andrieux, Simon Lagies, Johanna Bartmuss, Anie Masilamani, Alix Wasilenko, Daniela Osti, Stefania Faletti, Rana Schulzki, Shuai Yuan, Eva Kling, Valentino Ribecco, Henrik Heiland, Stefan Tholen, Marco Prinz, Giuliana Pelicci, Bernd Kammerer, Melanie Börries, and Maria Stella Carro

DOI: <https://doi.org/10.26508/lsa.202201400>

Corresponding author(s): Maria Stella Carro, University Medical Center Freiburg

Review Timeline:

Submission Date:	2022-02-04
Editorial Decision:	2022-03-29
Revision Received:	2022-09-27
Editorial Decision:	2022-10-14
Revision Received:	2022-10-28
Editorial Decision:	2022-10-28
Revision Received:	2022-10-28
Accepted:	2022-11-02

Scientific Editor: Novella Guidi

Transaction Report:

March 29, 2022

Re: Life Science Alliance manuscript #LSA-2022-01400-T

Dr. Maria Stella Carro
University Medical Center Freiburg
Germany

Dear Dr. Carro,

Thank you for submitting your manuscript entitled "ZBTB18 inhibits SREBP-dependent lipid synthesis by halting CTBPs and LSD1 activity in glioblastoma" to Life Science Alliance. The manuscript was assessed by expert reviewers, whose comments are appended to this letter. We, thus, encourage you to submit a revised version of the manuscript back to LSA that responds to all of the reviewers' points.

Thank you for this interesting contribution to Life Science Alliance. We are looking forward to receiving your revised manuscript.

Sincerely,

B. MANUSCRIPT ORGANIZATION AND FORMATTING:

Reviewer #1 (Comments to the Authors (Required)):

In this manuscript, a complex formed by ZBTB18, CtBP and LSD1 is proposed to regulate SREBF1 gene expression and fatty acid metabolism in glioblastoma cells. While the observation in this manuscript is of interest, some of the conclusions are not well supported by the experimental data especially for the lack of rescue experiments and phenotype validation. Therefore, this work needs a significant improvement for the data presentation through which some solid conclusions can be made. For details, please see below.

Major concerns:

1. In figure 1, there is no signal from ZBTB18, CtBP and Flag-ZBTB18 mutation in input data of Figure A, B, C and E. The input only counts 2% of the total extract, the author might need to increase the input or extend the exposure time. Appearance of control is critical to be used as the reference for explain the other signals. Similar problem happened to Fig2K where the control band is invisible or too weak. In fact, Fig2K is sgRNA mediated KO, the sequencing data of the KO gene loci should be provided.
2. In figure 2 (L) The graph color of SCD should be revised as you labeled. P value should be provided as the data seems not significant. If the data is not significant, the author should explain the reason. IN addition, based on the RT-PCR work shown in Figure 2 L, KO ZBTB18 shows a limited influence on SCD, LDLR, FASN, SREBF1 etc, but in Fig3G the phenotype shows a very dramatically change upon sgZBTB18. Is there any explanation for this inconsistency? If yes, more experiments to rescue the phenotype shown in Fig3G should be performed.
3. Fig4B, the labels for the overlaps are not understandable. For instance, the center portion is labeled as "All (5361)". How this number is derived?
Fig4I, ZBTB18 expression increases CtBP binding at Fatty acid metabolic genes promoters. Gene expression data already showed in Fig2H and 2J that CtBP increases while ZBTB18 decreases these gene expression. Could the author explain why ZBTB18 need to recruit CtBP to repress these gene expression while CtBP alone is considered as a gene activator?
Fig4E in "only CtBP2_ZBTB18 group", no ZBTB18 motif was find. need explains.
4. The interaction between ZBTB18 and ZNF217-CtBP-LSD1 has been partially proved in this manuscript. But there is no experiment to confirm the association between the fatty acid metabolism and the binding activity of these factors at the target gene promoters. The dependence of fatty acid metabolism gene specific ZNF217-CtBP-LSD1 complex recruitment on ZBTB18 should be confirmed. Also, whether the ZNF217 DNA sequence specific binding activity is required in this fatty acid gene regulation should be explained. Moreover, the significance of this complex in regulating faty acid metabolism should be validated by loss-of function studies coupled with rescue experiments.
3. In Figure 6, the manuscript showed that LSD1 is a activator of fatty acid metabolism genes as CtBP. However, the data has not been connected to ZBTB18 by either the change of gene expression or binding of these regulatory factors. The functional connection, as well as the regulatory connection between these two factors should be addressed experimentally. The same problem applies to other factors like ZNF217.
Another question is that can you explain why the treatment of RN-1 will significantly increase the pull-down efficiency of LSD1 (in IP LSD1 group of Figure 6 J) and CtBP2 (in IP CtBP2 group of Figure 6 K).
4. Previously study has proved that CtBP negatively regulates SREBF and promotes metastasis of breast cancer (pubmed, 30442980). The conclusion in this manuscript shows an opposite effect for CtBP2 in glioblastoma. Please discuss the discrepancy.

Minor:

IN fig3E and 3F, n=4 was provided in legend. Please provide the detail procedure for acquiring the signal as shown in the histogram.

Fig4D, if the TSS is SREBP genes, the Y-axis is indicated as percentage of peaks, what is the meaning of this "percentage"? For a defined genomic region, the true reads mapping might be better to show the signal strength.

Fig4G, SREBF genes should be included together.

Reviewer #2 (Comments to the Authors (Required)):

In this manuscript, Ferrarese and co-authors sought to examine the role of ZBTB18, a newly identified tumor suppressor with low expression in GBM cell lines. Their results define a model wherein ZBTB18 levels determines fatty acid synthesis in GBM cells. Low expression of ZBTB18 in glioma cells allows CTBP and LSD1 complex binding to the promoter regions of SREBP dependent genes, allowing gene expression. They then show that ectopic ZBTB18 expression in these cells, promotes the formation of a complex involving ZBTB18, CTBP1/2 and LSD1, a process facilitated by the presence of CTBP binding sequences in ZBTB18 protein. Further, they show that LSD1 acts as a scaffolding protein, rather than the "canonical" demethylase when in the context of this complex. This therefore, prevents the activation of gene expression and consequently leads to a reduction in the levels of phospholipids, specifically those containing unsaturated fatty acids. Overall, the study provides a new insight into the mechanism by which glioma cells maintain high levels of fatty acid synthesis and offers a mechanism that could downregulate this pathway. The manuscript is written clearly. However, addressing certain specific comments stated below will greatly enhance the clarity of the manuscript:

Major comments:

Figure 2H: "When MTOB treatment was combined with ZBTB18 or ZBTB18-mut expression in BTSC233, no rescue of ZBTB18 repression was observed (Figure 2H); however, ZBTB18-mut seemed to be less effective in repressing SREBP genes (Figure 2H), suggesting that MTOB might specifically affect the transcriptional activating role of CTBP without impairing ZBTB18 repressive function. Consistent with this idea, MTOB was shown to cause displacement of CTBP from its target gene promoters with no impairment of its interaction with transcription factors (Di et al. 2013)"

This line of reasoning seems to be arbitrary and vague. Firstly, given that CTBP lies downstream of ZBTB18 which represses it, one would not expect any rescue of ZBTB18 OE upon adding MTOB. Rather a further downregulation is to be expected as seen in Figure 2E using CTBP KD. Further, it seems that MTOB treatment should be able to downregulate gene expression even when added on ZBTB18 mut expressing cells. This is not the case for several genes (LDLR, SCD, GPAM). What is the explanation for this?

Figure 2L: The gene expression shows only a very small increase, which is not seen in both guide RNA constructs. Therefore, this is not convincing. The likely reason for this is the very low levels of ZBTB18 to begin with. Therefore, it is not unexpected that a mild decrease in protein levels has no major effect.

Figure 3: While it is clear that ZBTB18 OE indeed decreased lipid droplet content without affecting palmitate uptake that in itself does not conclusively prove that ZBTB18 affects de novo lipogenesis. To clearly demonstrate this, the authors should perform a glucose labelling experiment and determine if indeed incorporation of glucose derived carbon into fatty acids is affected. Further, since the authors claim that desaturation is particularly impacted, the glucose labelling will help examine if the effect is mainly on desaturation steps or on fatty acid synthesis as well. This is important because fatty acid synthetase as well as desaturases are SREBP targets.

Figure 5G-I: It seems clear that ZBTB18 regulates H3K4me2 and H3K9me2 at specific promoters. However, it would be important to determine if ZBTB18 regulates global methylation levels as well. Particularly because LSD1 is a global demethylase and regulates methylation at several gene promoters. Checking global methylation levels will be important to understand if ZBTB18 regulates gene specific methylation or has a rather global effect.

Figure 6H-I: RN1 treatment increasing H3K4me2 and H3K9me2 is to be expected. It would be interesting to see the levels upon RN1 treatment in ZBTB18 overexpressing cells. As ZBTB18 reduces the canonical demethylase activity of LSD1 (Figure 5-supplement 1E), it is possible that RN1 has no effect in OE cells. This would suggest a predominantly scaffolding function of LSD1 when ZBTB1 is present and therefore solidify the authors' claim of altered LSD1 activity when in a complex with ZBTB18.

General comments:

-It seems that GSCs having low basal ZBTB18 levels allows high expression of lipid metabolism genes. Therefore, what is the phenotypic consequence of ZBTB18 expression in terms of proliferative or invasive capacity and chemo sensitivity?

-The Figure representation should be changed to increase clarity:

In several Figures such as Figure 1G, 2F the y-axis is fold change. It should be indicate what kind of fold change e.g. mRNA expression fold change...

In several Figures such as Figure 2H, J, L the y-axis is arbitrary units thus a normalized value should be presented instead of y-axis values of 10^{-5} .

In Figure 2L it is not clear what is the difference between the three red bars for SCD

Why is the %input ranging from 0.05 to 3 % across multiple Figures?

For Figure 5D error bars and statistics should be added

For each bar graph single measurements need to be represented.

Minor comments:

Abstract: "Our study points at CTBP1/2 and LSD1 as co-activators of SREBP genes whose complex functional activity is altered by ZBTB18"

Rephrase to make it clear that the "functional activity of the complex" is being referenced.

Figure 1C: Why is there no CTBP2 protein detected in the input sample?

Figure 2K: The confirmation blot for ZBTB18sg#1 and #2 can be removed, as they have not been used in the study.

Figure 3-figure supplement 1 A-B: The y-axis label should be corrected.

Discussion: "It has been previously reported that malignant glioma tissues contain high levels of polyunsaturated fatty acids (Srivastava et al. 2010). Since the SREBF1 target SCD converts newly synthesized saturated fatty acids to unsaturated fatty acids (Cohen et al.

2002), it is possible that ZBTB18-mediated SCD repression accounts for the observed reduction of unsaturated and polyunsaturated fatty acids"

-This statement should be revised, as SCD is not involved in synthesis of polyunsaturated Fatty acids. Therefore, a repression of SCD cannot directly affect PUFA synthesis.

General: Throughout the article, the authors have used several different cell lines for their experiments, without offering an explanation for switching. While they are all GBM stem cell lines, it would be important to clarify why the experiments were not consistently performed in the same cell line.

Reviewer #3 (Comments to the Authors (Required)):

Manuscript: ZBTB18 inhibits SREBP-dependent fatty acid synthesis by counteracting CTBPs and KDM1A/LSD1 activity in glioblastoma

Authors: Ferrarese R.1, Izzo A.1, Andrieux G.2,3, Lagies S.4,5,6, Bartmuss J.P.1, Masilamani A.P.1, Wasilenko A.1, Osti D.7, Faletti S.7, Schulzki R.1, Shuai Y. 1, Kling E.1, Ribocco V.1, Heiland D.H.1, Tholen S.G.8, Prinz M. 9,10,11, Pelicci G.7,12, Kammerer B.4,5,13, Bőrries M.2,3 and Carro M.S.1*

Summary and comments:

Establishment of key mechanisms underlying dysregulated lipid metabolism is crucial for a better understanding of metabolic abnormalities in cancer. The present study explores the role of ZBTB18 in the regulation of cholesterol and fatty acids synthesis through transcriptional regulation of SREBPs, in Glioblastoma. Using Co-immunoprecipitation, mass spectrometry, ChIP-seq, shRNA, and gene expression profiling, the authors showed that ZBTB18 may repress the expression of SREBP genes in a CTBP-dependent manner. They demonstrated that ectopic ZBTB18 binds to SREBP promoter regions and contracts the CTBP/LSD1 activity to repress SREBP. Consequently, overexpression of ZBTB18 was associated with reduced lipid synthesis and storage. Overall, the paper is well-written and organized. The finding is interesting and makes an important contribution to cancer metabolism research. However, there are some methodological issues and few concerns with this study. Specific comments follow

Although the study was smoothly designed to explore potential interactions upon ZBTB18 overexpression and support previous findings, A major drawback of this study is the lack of controls, and the use of single cell line (SNB19) for most of the experiments. It appears that controls are missing in some Western Blots. In shRNA silencing experiment, only a single shRNA was used. Additionally, it would have been more rational if the authors used various GBM cell lines to better connect these findings to GBM biology.

The authors demonstrated that ZBTB18 loss in BTSC475 cell line enhanced lipid droplet accumulation with a modest repression of SREBP (SREBF1). These results were not well developed despite their importance and relevance to the interactions being studied thereafter. This raises another relevant question: why the authors did not investigate CTBP2 interaction with endogenous ZBTB18 in GBM cell lines expressing the later, in tandem with the ZBTB18 overexpression?

Most importantly, it would have been more reasonable to first consider showing the basal expression of ZBTB18 in GBM cell line being used, as well as other non-glioma cell lines as controls (which are lacking). Then, a design with KD or Loss of function of ZBTB18 in GBM and control cell lines should be considered in all experiments. This would further elucidate the effect of ZBTB18 on the interactions between the other regulators : CTBP/LSD1/ZNF217, their recruitment to SREBP promoter, as well as the expression of SREBP and the lipidomic profile of the cells. Such experiment will give more insights whether ZBTB18 suppression exacerbate the phenotype associated with enhanced FA synthesis in GBM, and in the control cell lines as well . In the proposed model, the role of ZBTB18 with respect to SREBP co-activators, particularly LSD1 and ZNF217, is not completely clear. While ZBTB18 acts as a repressor of SREBP, the SILAC/IP results simultaneously show that ZBTB18 reinforces the interaction between CTBP, LSD1, and ZNF217, which are co-activators of SREBP. Yet, the level of the co-activator LSD1 was increased upon expression of ZBTB18. This appears to be inconsistent and may require further discussion and clarification to support and refine the overall model.

Material and methods section. information concerning shRNA silencing and CRISPR/Cas9 experiments (Construct; LV production, and transduction) is missing. I would suggest that the authors describe these methods. Similarly, analysis of ChIP-seq data are not well reported and needs further development on how data was processed and filtered .

Strangely, the ChIP-seq dataset generated by this study is not found on GEO using the accession number provided in the manuscript (GSE140002). such data should be made publicly available for review and further analysis.

Statistical analysis. Data in some figures are plotted without any statistical analysis/comparison (e.g., 4 H-I, 5 D-I, 6 H-I) Fig 1.B the legend needs to be completed. which CTBP (1 or2) is shown in the figure? and why are there two distinct bands for ZBTB18 ?

Reviewer #1 (Comments to the Authors (Required)):

In this manuscript, a complex formed by ZBTB18, CtBP and LSD1 is proposed to regulate SREBF1 gene expression and fatty acid metabolism in glioblastoma cells. While the observation in this manuscript is of interest, some of the conclusions are not well supported by the experimental data especially for the lack of rescue experiments and phenotype validation. Therefore, this work needs a significant improvement for the data presentation through which some solid conclusions can be made. For details, please see below.

Major concerns:

1. In figure 1, there is no signal from ZBTB18, CtBP and Flag-ZBTB18 mutation in input data of Figure A, B, C and E. The input only counts 2% of the total extract, the author might need to increase the input or extend the exposure time. Appearance of control is critical to be used as the reference for explain the other signals. Similar problem happened to Fig2K where the control band is invisible or too weak. In fact, Fig2K is sgRNA mediated KO, the sequencing data of the KO gene loci should be provided.

Answer:

We agree with the Reviewer that it would be important to detect ZBTB18 in the input. However, due to the low expression of endogenous ZBTB18 in BTSC cell lines, as shown in our previous studies (Fedele et al., 2017 and Masilamani et al., 2022), it is not easy to detect it in the input, which represents 2% of the lysate used for IP. Increasing the input is not a feasible option since we cannot load more than 2-5% of the lysate used for IP. The exposure time was already increased but sometimes it was not sufficient to detect the endogenous ZBTB18. In Fig.1A-B and E, CTBP1/2 signal is visible, although weak. We have included a new image with higher exposure (Fig. 1E). In Fig.S1, we have included a new western blot to confirm that the CTBP band detected in the immune-precipitated fraction corresponds to the one detected in the input. To further support our conclusion, in the revised Fig. 1 we have included co-IP performed in SNB19 cells transduced with ZBTB18 (Fig 1A-B). We have moved Fig 1B in Fig S1D. Overall, considering that CTBP1/2 is detected as ZBTB18 interactor in all the coIP-MS analysis (Fig. S1), we are confident about the outcome of validation experiments shown in Fig.1.

In Figure 2K (now Fig. S6B), the Ctr ZBTB18 band (doublet) is visible, although faint due to the low expression level of ZBTB18 in this cell type. As mentioned above, as well as in the manuscript and our previous studies (Fedele et al., 2017 and Masilamani et al., 2022), ZBTB18 expression is low or absent in GBM-derived cells. Patient-derived BTSC475 were chosen to establish ZBTB18 knockout since they express a basal level of ZBTB18, detectable by western blot. In fact, it is present in sgRNA#1 and #2 but is absent in samples sgRNA#3 and #4. We have confirmed that the *ZBTB18* gene locus has been altered by CRISPR/Cas9 and have now included the sequencing data in the revised manuscript (Fig. S6A). Both guide RNAs led to the introduction of one or two additional nucleotides, which caused a frame shift.

2. In figure 2 (L) The graph color of SCD should be revised as you labeled. P value should be provided as the data seems not significant. If the data is not significant, the author should explain the reason. IN addition, based on the RT-PCR work shown in Figure 2 L, KO ZBTB18 shows a limited influence on SCD, LDLR, FASN, SREBF1 etc, but in Fig3G the phenotype shows a very dramatically change upon sgZBTB18. Is there any explanation for this inconsistency? If yes, more experiments to rescue the phenotype shown in Fig3G should be performed.

Answer:

We thank the Reviewer for noticing this mistake. The SCD color is now correct (Fig S6C). Statistical significance is represented with asterisk code whenever present. We agree with the Reviewer that the ZBTB18 KO seems to have a limited influence. It must be mentioned that the change in LD abundance is quantified in Fig. 3H. Pictures in 3G were selected based on their measured fluorescence intensity (as close as possible to the average) and on having a similar cell density. We have now selected a different image for the control sample, which better represents the overall result. It is worth noticing that while the function of ZBTB18 exerts an arguably direct effect on the gene expression, the accumulation of lipid droplets is a dynamic phenomenon depending on several factors, which ultimately determine their turnover. The rationale behind the lipid droplets quantification upon ZBTB18 KO was to compare it with the previously collected data when ZBTB18 was ectopically expressed in GBM cells. In this context, we believe LD abundance represents a relevant marker of ZBTB18 activity. We agree with the Reviewer that the result of this experiment is surprisingly sharp, especially if compared with the modest SREB gene expression changes. However as mentioned above, LD differential accumulation might rely on parameters we did not measure in our experiments. Overall, the knockdown experiments have been included as a proof of principle to show that removing ZBTB18 in GBM cells causes opposite effects compared to the overexpression studies. On the other hand, we realize that the level of ZBTB18 expression in all the BTSCs analyzed so far is low (Fedele et al., 2017 and Masilamani et al., 2022); thus, in agreement with Rev. #2, we acknowledge that BTSCs are not a very suitable standalone model to study the effect of ZBTB18 knockdown in GBM cells.

3. Fig4B, the labels for the overlaps are not understandable. For instance, the center portion is labeled as "All (5361)". How this number is derived?

Answer:

As mentioned in the text (page 11 of the revised manuscript) this number indicates “the number of peaks shared by all the conditions” (i.e., EV_ChIP: CTBP2, ZBTB18_ChIP: CTBP2, and ZBTB18_ChIP: FLAG, after subtracting FLAG_unspecific peaks detected in EV ChIP-FLAG). EV means “Empty vector”. We have modified the text both in the Results and in the Figure Legends (Fig 4A-B) sections to clarify the content of each group.

Fig4I, ZBTB18 expression increases CtBP binding at Fatty acid metabolic genes promoters. Gene expression data already showed in Fig 2H and 2J that CtBP increases while ZBTB18 decreases these gene expression. Could the author explain why ZBTB18 need to recruit CtBP to repress these gene expression while CtBP alone is considered as a gene activator?

Answer:

According to our model when ZBTB18 expression is absent or low, CTBP along with LSD1 behaves as transcriptional activator of SREBP genes in GBM cells. We propose that upon overexpression, ZBTB18 binds to the complex by direct interaction with CTBP and inhibits its activating function, including the inhibition of LSD1 demethylase activity. In addition, the data hint to the possibility that the presence of ZBTB18 in the CTBP-LSD1 complex might promote and/or stabilize the interaction with other repressive co-factors, such as ZNF217, likely through a conformational change of LSD1, which could lose the demethylase activity in favor of a more prominent scaffolding function. The Co-IP in Fig. 5D suggests that LSD1 protein could in fact be stabilized. Therefore, we propose that ZBTB18 presence modifies the composition and the enzymatic activity of the CTBP-LSD1 complex by dampening its demethylase activity and promoting LSD1 scaffolding function. However, the exact contribution of LSD1 scaffolding to the recruitment of additional co-factors and to the repression of SREBP genes needs further studies to be confirmed and fully characterized.

Fig4E in "only CtBP2_ZBTB18 group", no ZBTB18 motif was found. need explains.

Answer:

As we mentioned in the text (page 12 of the revised manuscript), we think that promoter peaks belonging to the group "Not ZBTB18_CTBP2" could be unspecific. These peaks are not shared by ZBTB18 and CTBP2 upon ZBTB18 expression and do not contain the ZBTB18 motif.

4. The interaction between ZBTB18 and ZNF217-CTBP-LSD1 has been partially proved in this manuscript. But there is no experiment to confirm the association between the fatty acid metabolism and the binding activity of these factors at the target gene promoters. The dependence of fatty acid metabolism gene specific ZNF217-CtBP-LSD1 complex recruitment on ZBTB18 should be confirmed.

Also, whether the ZNF217 DNA sequence specific binding activity is required in this fatty acid gene regulation should be explained. Moreover, the significance of this complex in regulating fatty acid metabolism should be validated by loss-of function studies coupled with rescue experiments.

Answer:

The Reviewer correctly points out that some components of the newly identified complex have not been strongly characterized in the context of fatty acid regulation (i.e., LSD1 and ZNF217). While we acknowledge that it would be interesting to define the role of all the complex subunits, we also believe that investigating the function of each component would require a significant amount of additional experiments and extend the paper beyond its original scope. Nevertheless, we have performed new experiments to clarify the role of LSD1 as a regulator of SREBP genes and fatty acid synthesis in presence and in absence of ZBTB18 (Author response figure 1). Both SREBP gene expression analysis by qPCR and ¹³C tracing experiments have shown that depletion of LSD1 in GBM cells leads to SREBP gene downregulation and reduction of *de novo* fatty acid synthesis. These data are now shown in Fig 6D-E. Similarly to what we observed upon CTBP knockdown or inhibition with MTOB, ZBTB18 knockdown does not rescue LSD1 KO phenotype; this is consistent with the idea that ZBTB18 acts by repressing LSD1. Furthermore, our data suggest that the interaction of ZBTB18 with the CTBP-LSD1 complex might also promote LSD1 scaffolding function. However as previously mentioned, this will require further studies.

Author response figure. 1. Top panel: qPCR showing expression levels of SREBF1 and SCD genes upon LSD1 knockout in presence or absence of ectopic ZBTB18. Bottom panel: Analysis of the ¹³C-labelled glucose incorporation into fatty acids in GBM#22 cells upon CRISPR/Cas9-mediated LSD1 knockdown (sgLSD1#1) and transduced with empty vector (EV) or ZBTB18 overexpressing vector (ZBTB18). The three principal lipid species emerged from the analysis are individually represented to show the differential progressive incorporation of ¹³C.

5. In Figure 6, the manuscript showed that LSD1 is an activator of fatty acid metabolism genes as CtBP. However, the data has not been connected to ZBTB18 by either the change of gene expression or binding of these regulatory factors. The functional connection, as well as the regulatory connection between these two factors should be addressed experimentally. The same problem applies to other factors like ZNF217.

Answer:

While we think that the characterization of the role of ZNF217 is not the major focus of our study (please, see response above), we agree that it would be important to complete Figure 6, connecting the role of LSD1 and ZBTB18 by gene expression change and chromatin binding. As reported above, qPCR experiments performed overexpressing ZBTB18 and knocking LSD1 down, indicate that LSD1 depletion does not impair ZBTB18-mediated repression of SREBP genes (Author response figure 1). Similar to what was observed upon CTBP2 knockdown, LSD1 silencing reduces SREBP gene activation, suggesting it works as an activator. It is worth mentioning that CTBP and/or LSD1 knockdown inhibit SREBP gene expression independently from the presence of ZBTB18, suggesting that the latter exerts its repressive function by halting the activating CTBP-LSD1 complex. This reinforces the hypothesis that ZBTB18 interaction with CTBP is required to inhibit CTBP and LSD1-mediated activation. However, we have also identified additional putative co-factors (such as ZNF217) that could have a yet-to-be-characterized role in ZBTB18-mediated repression of the SREBP genes. We intend to further investigate this aspect in a follow up study, focusing on the role of the different subunits of the ZBTB18 repressive complex.

Another question is that can you explain why the treatment of RN-1 will significantly increase the pull-down efficiency of LSD1 (in IP LSD1 group of Figure 6 J) and CtBP2 (in IP CtBP2 group of Figure 6 K).

Answer:

Although we do not know the exact mechanism, we hypothesize that inhibition of LSD1 demethylase activity through RN1 could induce structural changes in LSD1 that could affect its stability and interaction with other factors (i.e., CTBP2 and ZNF217). Similarly, upon ZBTB18 overexpression LSD1 demethylase activity is inhibited while LSD1 seems to be stabilized and to interact more with ZNF217 and CTBP. Consistent with this hypothesis, it has been shown that treatment with the SP2509 inhibitor, which impairs LSD1 interaction with ZNF217, reduces LSD1 half-life (Sehrawat et al., 2018), further indicating that LSD1 interaction with other cofactors is important for its stability.

6. Previously study has proved that CtBP negatively regulates SREBF and promotes metastasis of breast cancer (pubmed, 30442980). The conclusion in this manuscript shows an opposite effect for CtBP2 in glioblastoma. Please discuss the discrepancy.

Answer:

We are aware of the study mentioned by the Reviewer and have now commented on it in the discussion of the manuscript. We think that the main reason for this discrepancy lies in the different cell systems. According to our model, CTBP2 can be engaged in SREBP gene activation (in absence of ZBTB18) or repression (when it interacts with ZBTB18). Therefore, it is possible that in breast cancer CTBP2 could interact with ZBTB18 and repress SREBP genes. In addition, this mechanism may be dependent on CTBP2 interaction with ZEB1, in line with this possibility, ZEB1 was never detected in our CTBP2 co-immunoprecipitation analysis in GBM cells.

Minor:

IN fig3E and 3F, n=4 was provided in legend. Please provide the detail procedure for acquiring the signal as shown in the histogram.

Answer:

Five images for each of the four biological replicates were acquired with a confocal microscope. For each image, the average red channel intensity was measured with Photoshop, excluding black areas with no cells, when necessary.

Fig4D, if the TSS is SREBP genes, the Y-axis is indicated as percentage of peaks, what is the meaning of this "percentage"? For a defined genomic region, the true reads mapping might be better to show the signal strength.

Answer:

We thank the reviewer for his comment and we have modified the text and figures to clarify our findings. Briefly, the y-axis depicts the percentage of EV CTBP2, ZBTB18 CTBP2 and ZBTB18 FLAG peaks that overlap SREBP genes TSS regions. This percentage seems indeed rather low due to the high number of peaks we observed in total but in this way, we could normalize the overlap to the total number of peaks in each condition (i.e., EV CTBP2, ZBTB18 CTBP2 and ZBTB18 FLAG). Furthermore, we performed the same overlap analysis on randomly selected genes, and used an empirical cumulative distribution function to determine whether the observed percentage in SREBP genes was significantly higher than what we could expect to find by chance (from 1000 random sets of regions). P values are shown in Fig S8. In addition, we also agree that showing the true reads mapping of these defined genomic regions is useful to show the signal strength and we actually did it for the selected SREBP genes on Fig 4G, in which the y-axis represents read coverage.

Fig4G, SREBF genes should be included together.

Answer:

We have included SREBF1 in Fig 4G.

Reviewer #2 (Comments to the Authors (Required)):

In this manuscript, Ferrarese and co-authors sought to examine the role of ZBTB18, a newly identified tumor suppressor with low expression in GBM cell lines. Their results define a model wherein ZBTB18 levels determines fatty acid synthesis in GBM cells. Low expression of ZBTB18 in glioma cells allows CTBP and LSD1 complex binding to the promoter regions of SREBP dependent genes, allowing gene expression. They then show that ectopic ZBTB18 expression in these cells, promotes the formation of a complex involving ZBTB18, CTBP1/2 and LSD1, a process facilitated by the presence of CTBP binding sequences in ZBTB18 protein. Further, they show that LSD1 acts as a scaffolding protein, rather than the "canonical" demethylase when in the context of this complex. This therefore, prevents the activation of gene expression and consequently leads to a reduction in the levels of phospholipids, specifically those containing unsaturated fatty acids. Overall, the study provides a new insight into the mechanism by which glioma cells maintain high levels of fatty acid synthesis and offers a mechanism that could downregulate this pathway. The manuscript is written clearly. However, addressing certain specific comments stated below will greatly enhance the clarity of the manuscript:

We thank the Reviewer for this encouraging comment and for clearly summarizing our findings.

Major comments:

Figure 2H: "When MTOB treatment was combined with ZBTB18 or ZBTB18-mut expression in BTSC233, no rescue of ZBTB18 repression was observed (Figure 2H); however, ZBTB18-mut seemed to be less effective in repressing SREBP genes (Figure 2H), suggesting that MTOB might specifically affect the transcriptional activating role of CTBP without impairing ZBTB18 repressive function. Consistent with this idea, MTOB was shown to cause displacement of CTBP from its target gene promoters with no impairment of its interaction with transcription factors (Di et al. 2013)"

This line of reasoning seems to be arbitrary and vague. Firstly, given that CTBP lies downstream of ZBTB18 which represses it, one would not expect any rescue of ZBTB18 OE upon adding MTOB. Rather a further downregulation is to be expected as seen in Figure 2E using CTBP KD. Further, it seems that MTOB treatment should be able to downregulate gene expression even when added on ZBTB18 mut expressing cells. This is not the case for several genes (LDLR, SCD, GPAM). What is the explanation for this?

Answer:

We thank the Reviewer for this constructive comment. We agree that a rescue by MTOB in presence of ZBTB18 is not expected, since MTOB inhibits CTBP1/2 and its activating function. Therefore, we expect further downregulation, as shown in Fig 2E upon shCTBP2. This was indeed the case for SREBF1, SQLE and SCD. We have changed the text of the manuscript accordingly. The newly analyzed data of Fig 2H show that as expected, MTOB treatment is effective also in presence of ZBTB18-mut; the effect is especially clear for SREBF1 gene expression. For other genes (FASN, GPAM, LSS, SQLE and SCD), the effect was less evident but in line with the repression induced by MTOB alone. SREBF1 is also the most efficiently repressed gene upon CTBP2 silencing (Fig 2J), suggesting that the mechanism of gene

activation by CTBP2 could vary at different gene promoters (i.e., via the interaction with different co-factors).

Figure 2L: The gene expression shows only a very small increase, which is not seen in both guide RNA constructs. Therefore, this is not convincing. The likely reason for this is the very low levels of ZBTB18 to begin with. Therefore, it is not unexpected that a mild decrease in protein levels has no major effect.

Answer:

We agree with the Reviewer that the effect of ZBTB18 KO is quite modest. As correctly pointed out, ZBTB18 levels in GBM cells are very low and a clear detection of protein reduction is difficult to measure, even in BTSC475, which expresses slightly more ZBTB18 compared to other cells (Masilamani et al., 2022). Consequently, clear gene expression changes are difficult to determine.

Figure 3: While it is clear that ZBTB18 OE indeed decreased lipid droplet content without affecting palmitate uptake that in itself does not conclusively prove that ZBTB18 affects de novo lipogenesis. To clearly demonstrate this, the authors should perform a glucose labelling experiment and determine if indeed incorporation of glucose derived carbon into fatty acids is affected. Further, since the authors claim that desaturation is particularly impacted, the glucose labelling will help examine if the effect is mainly on desaturation steps or on fatty acid synthesis as well. This is important because fatty acid synthetase as well as desaturases are SREBP targets.

Answer:

We thank the Reviewer for the useful suggestion. We have performed a glucose labeling experiment in BTSC168 upon ZBTB18 expression.

Author response figure 2. Analysis of the ^{13}C -labelled glucose incorporation into fatty acids in BTSC168 transduced with empty vector (EV) or ZBTB18 overexpressing vector (ZBTB18). The four principal lipid species emerged from the analysis are individually represented to show the differential progressive incorporation of ^{13}C

In line with the previous observations, the results show that ZBTB18 expression reduced the incorporation of glucose-derived ^{13}C in several lipid species (Author response figure 2). These new data confirm our hypothesis that ZBTB18 inhibits fatty acid and cholesterol synthesis by repressing SREBP genes. The new results have been included in Fig 3I.

Figure 5G-I: It seems clear that ZBTB18 regulates H3K4me2 and H3K9me2 at specific promoters. However, it would be important to determine if ZBTB18 regulates global methylation levels as well. Particularly because LSD1 is a global demethylase and regulates methylation at several gene promoters. Checking global methylation levels will be important to understand if ZBTB18 regulates gene specific methylation or has a rather global effect.

Answer:

We thank the Reviewer for this suggestion. However, we think that investigating the effect of ZBTB18 on global methylation, albeit certainly interesting, would require an extensive effort and would go beyond the current focus of our study. Therefore, we have given priority to experiments that in our opinion, were required to strengthen our model.

Figure 6H-I: RN1 treatment increasing H3K4me2 and H3K9me2 is to be expected. It would be interesting to see the levels upon RN1 treatment in ZBTB18 overexpressing cells. As ZBTB18 reduces the canonical demethylase activity of LSD1 (Figure 5-supplement 1E), it is possible that RN1 has no effect in OE cells. This would suggest a predominantly scaffolding function of LSD1 when ZBTB1 is present and therefore solidify the authors' claim of altered LSD1 activity when in a complex with ZBTB18.

Answer:

We have now included a new graph (Fig 5H and Author response figure 3) that shows the effect of combining ZBTB18 expression and LSD1 inhibition through RN1, on H3K4me2 and H3K9me2. As hypothesized by the Reviewer, we did not observe a further effect by RN1 in presence of ZBTB18, in agreement with the idea that LSD1 demethylase activity is already reduced in presence of ZBTB18.

Author response figure 3. Representative ChIP qPCR showing the enrichment of H3K4me2 and H3K9me2 marks in BTSC168, transduced with empty vector (EV), FLAG-ZBTB18 or FLAG-ZBTB18mut and treated with the LSD1 inhibitor RN1, at the *SREBF1*, *SCD* and *FASN* promoters.

General comments:

-It seems that GSCs having low basal ZBTB18 levels allows high expression of lipid metabolism genes. Therefore, what is the phenotypic consequence of ZBTB18 expression in terms of proliferative or invasive capacity and chemo sensitivity?

Answer:

In our previous manuscript, we have investigated the role of ZBTB18 in glioblastoma (Fedele et al., 2017). ZBTB18 reduces the proliferative and invasive abilities of GBM cells, as well as tumor formation *in vivo*. In this manuscript, in Fig S2 we confirm this phenotype and show that it partially depends on CTBP binding.

-The Figure representation should be changed to increase clarity:

In several Figures such as Figure 1G, 2F the y-axis is fold change. It should be indicate what kind of fold change e.g. mRNA expression fold change...

In several Figures such as Figure 2H, J, L the y-axis is arbitrary units thus a normalized value should be presented instead of y-axis values of 10^{-5} .

In Figure 2L it is not clear what is the difference between the three red bars for SCD
Why is the %input ranging from 0.05 to 3 % across multiple Figures?

Answer:

We have improved the figure representation as suggested by the Reviewer. Regarding the difference in the % input range in various ChIP experiments, this is due to the efficiency of the ChIP, which is mainly affected by the quality of the antibodies (ChIP grade). The percentage of immunoprecipitated factors is usually higher for histone marks (i.e., H3K4me2 and H3K9me2).

For Figure 5D error bars and statistics should be added

Answer:

The plot in Fig 5D referred to the band quantification of a representative experiment; no statistical analysis was applied to the quantification. However, since the change of the level of the examined proteins is clear, we have removed the quantification, in order to avoid confusion.

For each bar graph single measurements need to be represented.

Answer:

We have replaced all the plots and created new graphs including standard deviation and single measurements, as suggested.

Minor comments:

Abstract: "Our study points at CTBP1/2 and LSD1 as co-activators of SREBP genes whose complex functional activity is altered by ZBTB18"

Rephrase to make it clear that the "functional activity of the complex" is being referenced.

Answer:

We have rephrased the sentence as follows: "Our study identifies CTBP1/2 and LSD1 as co-activators of SREBP genes and indicates that the functional activity of the CTBP-LSD1 complex is altered by ZBTB18."

Figure 1C: Why is there no CTBP2 protein detected in the input sample?

Answer:

Since we loaded 2% of the input for reference, it is possible that this amount is not enough to detect the endogenous protein in this cell type (LN229). In order to confirm that the band observed in the precipitated fractions corresponds to CTBP2, we have included a new western blot (Fig S1C), which includes a total lysate sample of LN229 cells, as a reference.

Figure 2K: The confirmation blot for ZBTB18sg#1 and #2 can be removed, as they have not been used in the study.

Answer:

Although ZBTB18sg#1 and #2 have not been used in the study, we prefer to include the complete blot. In our opinion, this allows the reader to better appreciate the reduction of ZBTB18, which is quite faint in the parental cell line. The blot, along with the sequence and qPCR analysis, is now included in Fig S6.

Figure 3-figure supplement 1 A-B: The y-axis label should be corrected.

Answer:

We have corrected the figure (now Fig S7) as suggested.

Discussion: "It has been previously reported that malignant glioma tissues contain high levels of polyunsaturated fatty acids (Srivastava et al. 2010). Since the SREBF1 target SCD converts newly synthesized saturated fatty acids to unsaturated fatty acids (Cohen et al. 2002), it is possible that ZBTB18-mediated SCD repression accounts for the observed reduction of unsaturated and polyunsaturated fatty acids"

-This statement should be revised, as SCD is not involved in synthesis of polyunsaturated Fatty acids. Therefore, a repression of SCD cannot directly affect PUFA synthesis.

Answer:

We agree with the Reviewer. Our study was limited to SCD, which is responsible for the mono-unsaturation of fatty acid carbon chains. Even though it would be tempting to speculate on a possible role of SCD in prompting fatty acid poly-unsaturation and consequently, on a more prominent effect on the cellular phenotype, we do not have evidence to support the claim. Therefore, we decided to remove the statement the Reviewer is referring to entirely. It will be interesting to clarify this aspect in the future.

General: Throughout the article, the authors have used several different cell lines for their experiments, without offering an explanation for switching. While they are all GBM stem cell lines, it would be important to clarify why the experiments were not consistently performed in the same cell line.

Answer:

The majority of the experiments have been initially performed in the established GBM cell line SNB19, which grows in presence of serum. We then sought to validate our observations with primary GBM stem-like cells (BTSCs) grown in absence of serum with addition of growth factors, which represent a better cellular model for GBM. For the overexpression experiments, we mostly used BTSC233 and BTSC168, which do not express ZBTB18, while for the knockout studies we chose BTSC475, which expresses ZBTB18 at a basal level. A table, which contains info regarding ZBTB18 expression in BTSCs, has been included in our recently published article (Masilamani et al., 2022). We have now added a reference to this paper in the revised manuscript and tried to clarify the choice of the used cell lines.

Reviewer #3 (Comments to the Authors (Required)):

Manuscript: ZBTB18 inhibits SREBP-dependent fatty acid synthesis by counteracting CTBPs and KDM1A/LSD1 activity in glioblastoma

Authors: Ferrarese R.1 {section sign}, Izzo A.1 {section sign}, Andrieux G.2,3 {section sign}, Lagies S.4,5,6, Bartmuss J.P.1, Masilamani A.P.1, Wasilenko A.1 Osti D.7, Faletti S.7, Schulzki R.1, Shuai Y. 1, Kling E.1, Ribecco V.1, Heiland D.H.1, Tholen S.G.8, Prinz M. 9,10,11, Pelicci G.7,12, Kammerer B.4,5,13, Bőrries M.2,3 and Carro M.S.1*

Summary and comments:

Establishment of key mechanisms underlying dysregulated lipid metabolism is crucial for a better understanding of metabolic abnormalities in cancer. The present study explores the role of ZBTB18 in the regulation of cholesterol and fatty acids synthesis through transcriptional

regulation of SREBPs, in Glioblastoma. Using Co-immunoprecipitation, mass spectrometry, ChiP-seq, shRNA, and gene expression profiling, the authors showed that ZBTB18 may repress the expression of SREBP genes in a CTBP-dependent manner. They demonstrated that ectopic ZBTB18 binds to SREBP promoter regions and contracts the CTBP/LSD1 activity to repress SREBP. Consequently, overexpression of ZBTB18 was associated with reduced lipid synthesis and storage. Overall, the paper is well-written and organized. The finding is interesting and makes an important contribution to cancer metabolism research. However, there are some methodological issues and few concerns with this study. Specific comments follow

Although the study was smoothly designed to explore potential interactions upon ZBTB18 overexpression and support previous findings, A major drawback of this study is the lack of controls, and the use of single cell line (SNB19) for most of the experiments. It appears that controls are missing in some Western Blots. In shRNA silencing experiment, only a single shRNA was used. Additionally, it would have been more rational if the authors used various GBM cell lines to better connect these findings to GBM biology.

Answer:

We are glad that the Reviewer considers our study of interest. However, we kindly disagree with his criticism regarding the use of a single cell line and of a single shRNA. ZBTB18 overexpression experiments, as well as the CTBP silencing ones, have been performed at first in SNB19 and subsequently, they have been validated in several primary GBM stem-like cells (BTSCs). In addition, two independent silencing were used for CTBP1 and CTBP2, and two sgRNA were used for ZBTB18 knockdown. For LSD1, a single shRNA was used but the analysis was corroborated by additional experiments in which two independent sgRNA were used. Regarding the missing controls in western blot, in some co-immunoprecipitation experiments (i.e. Fig.1C) the corresponding band in the total is not visible, due to the low level of the endogenous protein. As a reference, we have now included the total lysates to the Co-IP experiments in Fig. S1C.

The authors demonstrated that ZBTB18 loss in BTSC475 cell line enhanced lipid droplet accumulation with a modest repression of SREBP (SREBF1). These results were not well developed despite their importance and relevance to the interactions being studied thereafter. This raises another relevant question: why the authors did not investigate CTBP2 interaction with endogenous ZBTB18 in GBM cell lines expressing the later, in tandem with the ZBTB18 overexpression?

Answer:

We agree with the Reviewer's comment. In fact, ZBTB18 interaction with CTBP1/2 was identified in BTSC268 and U3082, which express endogenous ZBTB18. Moreover, in Fig 1 ZBTB18 binding to CTBP1/2 was further validated by co-IP in BTSC268. However, knockdown experiments to study ZBTB18 function in GBM cells are not very feasible, considering the low expression of ZBTB18. As mentioned above, we have selected BTSC268 and BTSC475 as two cell lines with a detectable expression of ZBTB18. While BTSC268 cannot be used for the transfection, due to its resistance to the selective agent (puromycin), BTSC475 cells were chosen for ZBTB18 KO experiments aimed at corroborating our findings in the overexpression studies. As shown in Fig S6, ZBTB18 expression in BTSC475 is modest, therefore its knockdown elicited only a modest upregulation of the target genes. In the attempt to provide more studies with the KO tool, as requested, we have now performed ChIP experiments showing that ZBTB18 knockdown in BTSC475 is associated with a decrease of both H3K4me2 and H3K9me2 at the *LDLR* and *SCD* promoter, genes that were shown to be

upregulated at the gene expression level (Fig.2). The new data are shown below (Author response figure 4) and have been included in Fig 5I-J. Overall, as previously discussed, we concluded that given the very low level of ZBTB18 expression, BTSCs are not a suitable model to perform extensive knockdown experiments.

Author response figure 4. qPCR showing binding H3K4me2 H3K9me2 at the SREBF1 (I) and LDLR (J) gene promoters in BTSC475 cells transduced sgZBTB18 #3 and #4

Most importantly, it would have been more reasonable to first consider showing the basal expression of ZBTB18 in GBM cell line being used, as well as other non-glioma cell lines as controls (which are lacking). Then, a design with KD or Loss of function of ZBTB18 in GBM and control cell lines should be considered in all experiments. This would further elucidate the effect of ZBTB18 on the interactions between the other regulators:

Answer:

We agree with the Reviewer regarding the importance of showing a western blot analysis for comparison of ZBTB18 among different cell lines. The blots displayed below have been included in our recently published article, which has now been added as reference (Masilamani et al., 2022). Cortical tissues (normal brain, NB) are usually included as reference for ZBTB18

Author response figure 5. (A-B) Western blot analysis of ZBTB18 FL and SF expression in different GBM-derived primary cell lines (BTSCs) and one normal cortical tissue sample (NB), using an anti-ZBTB18 antibody

expression. Of note, ZBTB18 is not expressed in other potentially exploitable non-glioma cells, such as neural progenitors and astrocytes. As mentioned above, the level of ZBTB18 in the selected cell lines (BTSC268 and BTSC475), albeit higher compared to other BTSC cells, is anyway modest, limiting the possibility to manipulate its expression. In conclusion, we feel that our ZBTB18 knockdown experiments in BTSC475, even though not extensive and not conclusive *per se*, might support the results from the overexpression experiments and confirm the role of ZBTB18 as a negative regulator of fatty acid synthesis. We do agree that the model should be improved, especially concerning the possibility that upon inhibition of LSD1 demethylase activity by ZBTB18, the new scaffolding function of LSD1 might contribute to ZBTB18-mediated repression. In fact, we plan to follow up the project to investigate more in detail the role of CTBP/LSD1/ZNF217 complex and its association with endogenous ZBTB18.

CTBP/LSD1/ZNF217, their recruitment to SREBP promoter, as well as the expression of SREBP and the lipidomic profile of the cells. Such experiment will give more insights whether ZBTB18 suppression exacerbate the phenotype associated with enhanced FA synthesis in GBM, and in the control cell lines as well. In the proposed model, the role of ZBTB18 with respect to SREBP co-activators, particularly LSD1 and ZNF217, is not completely clear. While ZBTB18 acts as a repressor of SREBP, the SILAC/IP results simultaneously show that ZBTB18 reinforces the interaction between CTBP, LSD1, and ZNF217, which are co-activators of SREBP. Yet, the level of the co-activator LSD1 was increased upon expression of ZBTB18. This appears to be inconsistent and may require further discussion and clarification to support and refine the overall model.

Answer:

We agree with the Reviewer that the role of the CTBP/LSD1/ZNF217 complex is less characterized, in terms of association with the fatty acid synthesis in GBM. However, this mostly refers to the possibility that the CTBP/LSD1/ZNF217 complex recruited by ZBTB18 might further contribute to ZBTB18-mediated repression. As mentioned in the comments above, we plan to specifically investigate this possibility in a separate follow up study. In the current work, we propose that, in the absence of ZBTB18, CTBP and LSD1 function as activators of SREBP genes and that ZBTB18 inhibits their function. In the revised manuscript we have further reinforced this claim by showing that LSD1 knockdown causes a reduction of the synthesis of fatty acids (¹³C tracing experiment, included in Fig 6E). As shown in Fig 6D-E, ZBTB18 expression and LSD1 KO produce similar results, which are consistent with the idea that ZBTB18 impairs LSD1 (and CTBP) function (see also author response figure 2).

Regarding our model, we propose that ZBTB18, through the interaction with CTBP, inhibits CTBP-LSD1-mediated gene activation and favors the interaction of LSD1 and CTBP with ZNF217. Whether the stabilized/recruited CTBP-LSD1-ZNF217 complex has a co-repressive activity, which contributes to ZBTB18-mediated repression still needs to be clarified. In the revised manuscript, we have included a revised model (Fig 7) and discussion.

Material and methods section. information concerning shRNA silencing and CRISPR/Cas9 experiments (Construct; LV production, and transduction) is missing. I would suggest that the authors describe these methods. Similarly, analysis of ChIP-seq data are not well reported and needs further development on how data was processed and filtered.

Answer:

We are sorry that the Reviewer could not find this information, which was included in the Supplementary information file (supplementary methods section). These methods have been now moved to the Material and methods section.

Strangely, the ChIP-seq dataset generated by this study is not found on GEO using the accession number provided in the manuscript (GSE140002). such data should be made publicly available for review and further analysis.

Answer:

We apologize for that, we have included the token to access the data in the cover letter to the editor.

Statistical analysis. Data in some figures are plotted without any statistical analysis/comparison (e.g., 4 H-I, 5 D-I, 6 H-I)

Answer:

The statistical analysis is missing for the indicated ChIP, where a representative experiment is shown. We have indicated in the figure legends whether the average of multiple replicates or a representative experiment is shown.

Fig 1.B the legend needs to be completed. which CTBP (1 or2) is shown in the figure? and why are there two distinct bands for ZBTB18 ?

Answer:

In Figure 1B an anti-CTBP antibody that recognizes both CTBP1 and CTBP2 was used. We have explained it in the figure legend and added CTBP1/2 in the figure. The top ZBTB18 is likely a post-translationally modified form of ZBTB18. We have consistently detected these bands in our studies, including our recent manuscript (Masilamani et al., 2022). This panel has now been moved in Fig S1D.

October 14, 2022

Re: Life Science Alliance manuscript #LSA-2022-01400-TR

Dr. Maria Stella Carro
University Medical Center Freiburg
Breisacherstrasse 64
Freiburg 79106
Germany

Dear Dr. Carro,

Thank you for submitting your revised manuscript entitled "ZBTB18 inhibits SREBP-dependent lipid synthesis by halting CTBPs and LSD1 activity in glioblastoma" to Life Science Alliance. The manuscript has been seen by the original reviewers whose comments are appended below. While the reviewers continue to be overall positive about the work in terms of its suitability for Life Science Alliance, some important issues remain.

Our general policy is that papers are considered through only one revision cycle; however, given that the suggested changes are relatively minor, we are open to one additional short round of revision. Please note that I will expect to make a final decision without additional reviewer input upon resubmission.

Please submit the final revision within one month, along with a letter that includes a point by point response to the remaining reviewer comments.

To upload the revised version of your manuscript, please log in to your account: <https://lsa.msubmit.net/cgi-bin/main.plex>
You will be guided to complete the submission of your revised manuscript and to fill in all necessary information.

B. MANUSCRIPT ORGANIZATION AND FORMATTING:

Sincerely,

Reviewer #1 (Comments to the Authors (Required)):

The author has addressed most of my questions. Still some questions remain as listed below. In addition, the data presented so far still barely support the conclusions. The author should consider rephrasing their related conclusions, in particular when

speculate the working mechanism of how ZBTB18 forms complex with CTBP/LSD1/ZNF217.

Q1, the invisible input for ZBTB18 is not acceptable since the input sample should be the same sample as used for IP, it contains ectopically expressed ZBTB18. This has nothing to do with the endogenous ZBTB18.

In Fig1C, CtBP is not seen in input, not like the author declared as "visible".

Q3-2 Fig4I, the explanation is based on assumptions but not data supported. How the interaction between ZBTB18 and CtBP convert CtBP from an activator to repressor is not well supported by the data. In fact, in Fig2H, the SREBF1 gene is repressed by MTOB in combination with ZBTB18 or ZBTB18mut, this data argues against the proposed working model since MTOB could repress SREBF1 independent of ZBTB18. In addition, this data also shows that mutant ZBTB18 could repress SREBF1. If this is the case, how ZBTB18 mutant block CtBP/LSD1 activator function (as discussed in the rebuttal letter Q5 by the author)?

Reviewer #2 (Comments to the Authors (Required)):

The authors have improved the manuscript.

I am however puzzled about the new Figure 3I also Rev Figure 2.

This Figure shows the incorporation of ¹³C glucose into fatty acids. What is seen is a decrease in the y-axis but no shift in the x-axis for the label incorporation. The decrease in y-axis means that the fraction of newly synthesized fatty acids coming from glucose is reduced. The lack of a shift in x-axis means the total fraction of newly synthesis fatty acids is the same. The authors should do an ISA analysis to verify my interpretation (<https://doi.org/10.1016/bs.mie.2015.06.039>). Moreover, desaturation does not seem to be selectively effected. Thus, these data at this moment do not support the notion that fatty acid synthesis is effected, but rather the notion that the carbon source that is used for fatty acid synthesis is effect. The authors need to provide a correct interpretation of these data in the manuscript.

Reviewer #3 (Comments to the Authors (Required)):

Ferrarese and co-authors have appropriately, answered and addressed comments and concerns raised by the reviewers in the first review. I strongly believe that the manuscript is now much approved. The model proposed may provide new insight into the mechanism regulating fatty acids synthesis in Glioblastoma. However, minor additional comments listed Bellow should be addressed prior to acceptance.

- While the summary blurb is very useful, mentioning Glioblastoma may be more accurate in indicating the context of the finding.
- The role of CTBP2 in regulating SREBP expression conflicts with previously reported data, as mentioned in the manuscript. The authors should elaborate more and clarify how this might be context-depend, at the very least based on the recent literature, and what evidence support such differential function in Breast cancer, Unlike in the proposed model.

Reviewer #1 (Comments to the Authors (Required)): The author has addressed most of my questions. Still some questions remain as listed below. In addition, the data presented so far still barely support the conclusions. The author should consider rephrasing their related conclusions, in particular when speculate the working mechanism of how ZBTB18 forms complex with CTBP/LSD1/ZNF217.

Response: Through all the manuscript, we have clarified that ZBTB18 mediates SREBP gene repression by inhibiting CTBP- and LSD1-mediated activation. ZBTB18 expression, as well as LSD1 inhibition through treatment with RN1 inhibitor has an effect on CTBP/LSD1/ZNF217 interaction. We propose that inhibiting LSD1 demethylase activity could affect its scaffolding activity; however, we have clearly stated that this is a possible interpretation of the results and that more studies would be required to understand whether this complex further contributes to SREBP gene repression. To satisfy the reviewer's request, we have shortened the last paragraph of the introduction as follows: **“We report that CTBP and LSD1 transcriptionally activate the expression of fatty acid synthesis genes and that such activation is opposed by ZBTB18 through the inhibition of LSD1-dependent demethylase activity.”** Moreover, we have rephrased our conclusion as follows: **“In conclusion, we propose that ZBTB18 represses SREBP gene expression by inhibiting CTBP and LSD1-mediated transcriptional activation (Fig 7A-B). Moreover, ZBTB18 could promote LSD1 scaffolding function favoring the interaction between LSD1, CTBP and ZNF217 (Fig 7C).** In the future, more studies will be required to investigate whether the scaffolding role of LSD1 and its complex components (i.e., ZNF217) contribute to ZBTB18-mediated SREBP gene repression. Deciphering the precise molecular mechanism of fatty acid synthesis regulation will help define new potential therapeutic targets in GBM.”

In the discussion, the text has been changed as follows: **“Furthermore, our data indicate that, in presence of ZBTB18, LSD1 and CTBP more efficiently interact with each other and with the transcriptional co-repressor ZNF217. Similar results were obtained when LSD1 demethylase activity was inhibited by RN1. Therefore, we speculate that blocking LSD1 demethylase activity could favor LSD1 stability and scaffolding function. This hypothesis is consistent with recent studies indicating that LSD1 can repress gene expression independently from its histone demethylase activity (Sehrawat, Gao et al. 2018, Ravasio, Ceccacci et al. 2020). In particular, Sehrawat and colleagues demonstrated that LSD1 gene regulation in prostate cancer is mediated by interaction with ZNF217 independently of its demethylase activity. However, whether the scaffolding function of LSD1 and the recruitment of co-repressors such as ZNF217 further contribute to ZBTB18-mediated SREBP gene repression needs additional studies to be elucidated.”**

Q1, the invisible input for ZBTB18 is not acceptable since the input sample should be the same sample as used for IP, it contains ectopically expressed ZBTB18. This has nothing to do with the endogenous ZBTB18.

Response: It is not clear to which panel the reviewer is referring to. In figure 1A-B as well as in former Fig.1G, the ZBTB18 band (ectopically expressed) is visible in the input and is obviously missing in the empty vector samples. The CTBP2 band is faint but visible in Fig.1A-B and former Fig.1F. In former Fig. 1G, CTBP2 is not visible in the input but clearly enriched in the IP sample. It must be clarified that CTBP2 is not overexpressed and as an endogenous protein might not be sufficiently concentrated to be visible in the 2% input (in fact, we acknowledge that it is barely visible in Fig. 1F). In former Fig.1D, ZBTB18 (endogenous) is not visible since ZBTB18 expression is overall low. However also in this case, the band is clearly enriched in the IP sample. Moreover, CTBP and ZBTB18 interaction was detected in mass spec experiments using BTSCs that express endogenous ZBTB18 (revised Fig. S1E and S1G). Regarding former Fig. 1F-G, the input shown in the two panels comes from the same lysate, collected before the co-IP experiments. Since the CTBP2 and FLAG antibodies used are raised in different species (rabbit and mouse, respectively), the input samples were incubated

with different antibodies. So, although the FLAG antibody was not very efficient in recognizing the FLAG-ZBTB18 band in the input samples in former Fig. 1F, the western blot using anti-ZBTB18 antibodies clearly showed that ZBTB18 is expressed. Similarly, the CTBP2 antibodies used in the western blot in former Fig.1F (Active motif, rabbit) more efficiently detected the CTBP2 expression in the input (2%) compared to the CTBP2 antibody (BD, mouse) used in former Fig. 1G. To simplify the understanding of the experimental outcome, we have combined former Fig. 1F-G in a single panel (now Fig.1D), which shows the expression of ectopic ZBTB18 and endogenous CTBP2 in the input (2%).

Through the text, we have better described whether the co-IP experiments were performed in GBM cells that express endogenous or ectopic ZBTB18 and explained that due to the low fraction of the input loaded the endogenous proteins were not always visible in the input. Instead, to address the Reviewer's comment, all overexpressed proteins are now visible in the input (Fig. 1-B and Fig 1D).

In Fig1C, CtBP is not seen in input, not like the author declared as "visible".

Response: Yes, here the CTBP2 band in the input is not visible. For this reason, as explained, we added a western blot, which includes the total lysate as reference (new Fig. S1D). Since the co-IP experiment in LN229 cells adds up to other co-IPs shown in Fig.1, we have now moved it to Fig.S1C.

Q3-2 Fig4I, the explanation is based on assumptions but not data supported. How the interaction between ZBTB18 and CtBP convert CtBP from an activator to repressor is not well supported by the data. In fact, in Fig2H, the SREBF1 gene is repressed by MTOB in combination with ZBTB18 or ZBTB18mut, this data argues against the proposed working model since MTOB could repress SREBF1 independent of ZBTB18.

Response: The main conclusion of our study is that CTBP activating function is inhibited when ZBTB18 binds to CTBP itself; not that CTBP is turned into a repressor. Our data show that LSD1 and CTBP interaction with other cofactors (i.e., ZNF217) increases when also ZBTB18 is present, as a consequence of LSD1 activity inhibition. Prospectively, we propose that it would be important to investigate whether this complex (LSD1-CTBP-ZNF217) further contributes to ZBTB18-mediated repression. We have extensively mentioned this limitation in our previous point-by-point response as well as in the manuscript.

Regarding Figure 4I, in the text we only mention that "CTBP2 is present at the promoter of SREBP genes in the absence of ZBTB18 (Fig 4I) where according to our microarray analysis and qPCR results it is responsible for their transcriptional activation" without commenting on the fact that ZBTB18 could turn CTBP2 into a repressor. We do not imply that CTBP2 becomes a repressor upon ZBTB18 expression. We speculate that this is a possibility that should be investigated in the future. Furthermore, the criticism regarding MTOB is not correct. MTOB inhibits CTBP1/2, therefore we expect to observe SREBF1 downregulation, similarly to what was observed upon CTBP1/2 silencing and consistent with the idea that CTBP2 functions as an activator.

In addition, this data also shows that mutant ZBTB18 could repress SREBF1. If this is the case, how ZBTB18 mutant block CtBP/LSD1 activator function (as discussed in the rebuttal letter Q5 by the author)?

Response: In Fig.2H we show that ZBTB18-mut is in general unable to repress SREBP genes, or less efficient than ZBTB18 wild type. In particular, ZBTB18 overexpression or CTBP2 silencing lead to a clear downregulation of SREBF1 and the effect is even stronger when MTOB and ZBTB18 are used in combination. This is in line with our hypothesis that ZBTB18 functions by halting CTBP (and LSD1) activating function. However, we cannot exclude that ZBTB18 might also be able to repress gene expression independently from the binding of CTBP, as

discussed for the targets displayed in Fig.1H. We have realized that the paragraphs related to Fig. 2H were not clearly organized, which could have generated some confusion. In addition, we have added a more detailed comment regarding the different outcome of ZBTB18-mut expression, alone or combined with MTOB. We have modified the text as follows: “Treatment with MTOB in BTSC233 led to the downregulation of many SREBP genes, especially SREBF1 (Fig 2H). Of note, when MTOB treatment was combined with ZBTB18 or ZBTB18-mut expression, no rescue of ZBTB18-mediated repression was observed (Fig 2H). In fact, in some cases (SREBF1, GPAM, SQLE and SCD) the combined effect of MTOB and ZBTB18 appeared to be even stronger than the single treatments, similarly to what we had observed when ZBTB18 was expressed in concomitance with CTBP2 knockdown (Fig 2E). In presence of ZBTB18-mut, which has no significant effect on SREBF1, MTOB nonetheless elicited repression of the gene expression (Fig 2H); this is consistent with the idea that ZBTB18 and MTOB independently impair CTBP-mediated activation. However, although less efficient than ZBTB18 in repressing some of the tested SREBP genes, ZBTB18-mut still showed some degree of gene downregulation. This suggests that ZBTB18 may also be able to repress gene expression independent from CTBP binding, as previously observed (Fig.1H)”.

Reviewer #2 (Comments to the Authors (Required)): The authors have improved the manuscript. I am however puzzled about the new Figure 3I also Rev Figure 2. This Figure shows the incorporation of ¹³C glucose into fatty acids. What is seen is a decrease in the y-axis but no shift in the x-axis for the label incorporation. The decrease in y-axis means that the fraction of newly synthesized fatty acids coming from glucose is reduced. The lack of a shift in x-axis means the total fraction of newly synthesis fatty acids is the same. The authors should do an ISA analysis to verify my interpretation (<https://doi.org/10.1016/bs.mie.2015.06.039>). Moreover, desaturation does not seem to be selectively effected. Thus, these data at this moment do not support the notion that fatty acid synthesis is effected, but rather the notion that the carbon source that is used for fatty acid synthesis is effect. The authors need to provide a correct interpretation of these data in the manuscript.

Response: We agree with the Reviewer that the ¹³C-labelled glucose incorporation requires a more detailed interpretation. As suggested, an ISA analysis would indeed be required to confirm the Reviewer’s interpretation. Unfortunately, to do this we should set up an entirely new time-consuming experiment for which we lack the required expertise. It must be mentioned that we performed the experiment the Reviewer specifically requested in his comment in the previous round of revision. We performed the experiment with primary GBM stem-like cells (BTSC168) grown in complete BTSC medium, which contained glutamine, lipids and ¹³C-labelled glucose. So theoretically, other sources of carbon atoms (glutamine) or even already synthesized fatty acids (lipid uptake is not hindered by ZBTB18 expression) were available; for this reason, we cannot exclude that the reduction of ¹³C incorporation we observed was in fact limited to the tested carbon source (glucose), as correctly pointed out by the Reviewer. However, we believe this experiment should be interpreted in the context of the whole data presented in the manuscript: lipid synthesis seems affected by ZBTB18 from gene expression to lipid species abundance highlighted by lipidomics, to intracellular accumulation in lipid droplets. Regardless the ¹³C incorporation is limited to glucose only or rather it is a more general effect, ZBTB18 expression impacts *de novo* lipogenesis by somewhat reducing the rate to which carbon atoms are incorporated into novel fatty acid chains.

Thus, to provide a more accurate description of these results, we rephrased the manuscript as follows: “To further estimate the impact of ZBTB18 on *de novo* lipogenesis, we set up a ¹³C incorporation assay using labelled glucose in BTSC168 to quantitatively measure fatty acid synthesis. The results showed that, upon expression of ZBTB18, the detected fatty acids contained comparatively less glucose-derived ¹³C isotopes, suggesting a diminished request of carbon atoms for *de novo* lipogenesis (Fig 3I).”

“Consistent with the effect on SREBP gene expression, LSD1 depletion decreased glucose-derived ¹³C incorporation (Fig 6E).”

We have also added to the Discussion section what follows:

“The negative effect of ZBTB18 expression on lipogenesis is further supported by the reduced incorporation of glucose-derived ¹³C isotopes in newly synthesized fatty acids. Even though this assay is biased by the presence in the culture medium of other potential carbon atom sources (i.e., glutamine) and of lipids that can be directly taken up by the tumor cells, the results added up to the rest of the data, pointing to a role of the ZBTB18 complex in hampering *de novo* lipogenesis.”

Finally, the statements about fatty acid desaturation, which the Reviewer is referring to, had been already removed from the manuscript in the previous round of revision.

Reviewer #3 (Comments to the Authors (Required)): Ferrarese and co-authors have appropriately, answered and addressed comments and concerns raised by the reviewers in the first review. I strongly believe that the manuscript is now much approved. The model proposed may provide new insight into the mechanism regulating fatty acids synthesis in Glioblastoma. However, minor additional comments listed Bellow should be addressed prior to acceptance.

- While the summary blurb is very useful, mentioning Glioblastoma may be more accurate in indicating the context of the finding.

- The role of CTBP2 in regulating SREBP expression conflicts with previously reported data, as mentioned in the manuscript. The authors should elaborate more and clarify how this might be context-depend, at the very least based on the recent literature, and what evidence support such differential function in Breast cancer, Unlike in the proposed model.

Response: We are glad that the reviewer finds the manuscript much improved. As suggested, we have included “glioblastoma” in the summary blurb.

Regarding the role of CTBP2 in other contexts, we have included the reference to the Sekiya et al. study, which shows that CTBP2 represses fatty acid synthesis genes in the liver, through direct interaction with SREBF1, which is consequently inhibited. As mentioned further in the discussion, we think that the opposite role of CTBP as repressor or activator mostly depends on the role of its interacting partners and on the functional nature of the protein-protein interaction. In fact, while ZEB1 and CTBP2 cooperate to repress SREBF2 expression (Zhao, Hao et al. 2019), in the Sekiya et al. study CTBP2 binding to SREBF1 impairs SREBF1-mediated fatty acid gene expression. Conversely, in our model, CTBP function is linked to the activating role of LSD1.

We have modified the discussion as follows: “Recently, CTBP2 has been linked to the inhibition of cholesterol synthesis in breast cancer cells through direct repression of SREBF2 expression (Zhao, Hao et al. 2019). Furthermore, CTBP2 was shown to repress fatty acid biosynthesis in the liver (Sekiya, Kainoh et al. 2021). Although apparently in contrast with our proposed role of CTBP2 as activator of SREBP genes, it is possible that CTBP function might change based on the interaction with other transcription factors. According to this possibility, CTBP2 repression of fatty acid in breast cancer requires the interaction with the transcriptional repressor ZEB1, which was never detected as CTBP2 partner in our co-immunoprecipitation analysis in GBM cells. Instead, the study performed by Sekiya et al., proposes that CTBP2 interacts with SREBF1 protein to directly inhibit its transcriptional activity. However, these proposed models are mechanistically different from the one reported in our study, in which the newly described CTBP function seems to depend on the activating role of LSD1. This new role is consistent with previous studies showing that CTBP and LSD1 can be implicated in transcriptional activation (Ray, Li et al. 2014). It would be interesting to investigate whether other transcription factor participate to CTBP and LSD1-mediated SREBP gene activation.”

October 28, 2022

RE: Life Science Alliance Manuscript #LSA-2022-01400-TRR

Dr. Maria Stella Carro
University Medical Center Freiburg
Breisacherstrasse 64
Freiburg 79106
Germany

Dear Dr. Carro,

Thank you for submitting your revised manuscript entitled "ZBTB18 inhibits SREBP-dependent lipid synthesis by halting CTBPs and LSD1 activity in glioblastoma". We would be happy to publish your paper in Life Science Alliance pending final revisions necessary to meet our formatting guidelines.

- please use the [10 author names, et al.] format in your references (i.e. limit the author names to the first 10)
- please add your supplementary figure legends and your table legends to the main manuscript text; please upload your table files as editable doc or excel files or make sure that they are included in the doc file of your main manuscript text

A. FINAL FILES:

B. MANUSCRIPT ORGANIZATION AND FORMATTING:

**Submission of a paper that does not conform to Life Science Alliance guidelines will delay the acceptance of your

manuscript.**

The license to publish form must be signed before your manuscript can be sent to production. A link to the electronic license to publish form will be sent to the corresponding author only. Please take a moment to check your funder requirements.

Sincerely,

November 2, 2022

RE: Life Science Alliance Manuscript #LSA-2022-01400-TRRR

Dr. Maria Stella Carro
University Medical Center Freiburg
Breisacherstrasse 64
Freiburg 79106
Germany

Dear Dr. Carro,

Thank you for submitting your Research Article entitled "ZBTB18 inhibits SREBP-dependent lipid synthesis by halting CTBPs and LSD1 activity in glioblastoma". It is a pleasure to let you know that your manuscript is now accepted for publication in Life Science Alliance. Congratulations on this interesting work.

DISTRIBUTION OF MATERIALS:

Again, congratulations on a very nice paper. I hope you found the review process to be constructive and are pleased with how the manuscript was handled editorially. We look forward to future exciting submissions from your lab.

Sincerely,
